# Learning Personalized Decision Support Policies

## Abstract

Individual human decision-makers may benefit from different forms of support to improve decision outcomes, but *which* form of support will yield better outcomes? In this work, we propose the general problem of learning a *decision support policy* that, for a given input, chooses which form of support to provide to decision-makers for whom we initially have no prior information. Using techniques from stochastic contextual bandits, we introduce `THREAD`, an online algorithm to personalize a decision support policy for each decision-maker. We further propose a variant of `THREAD` for the multi-objective setting to account for auxiliary objectives like the cost of support. We find that `THREAD` can learn a personalized policy that outperforms offline policies, and, in the cost-aware setting, reduce the incurred cost with minimal degradation to performance. Our experiments include various realistic forms of support (e.g., expert consensus and predictions from a large language model) on vision and language tasks. We deploy `THREAD` with real users to show how personalized policies can be learned online and illustrate nuances of learning decision support policies in practice.

## 1 Introduction

To improve decision outcomes, human decision-makers can use various forms of support to inform their opinions before making a final decision (Keen, 1980). Decision-makers with differing expertise may benefit from different forms of support on a given input. For example, one radiologist may provide a better diagnosis of a chest X-ray by leveraging model predictions (Kahn Jr, 1994) while another may excel after viewing suggestions from senior radiologists (Briggs et al., 2008), or viewing a summary of relevant medical records from a large language model (LLM) (Yang et al., 2023a). In this paper, we study how to improve decision outcomes by *personalizing* which form of support we provide to a decision-maker on a case-by-case basis.

We formalize learning a *decision support policy* that dictates for each individual decision-maker what additional support, if any, should be presented for a given input (Figure 1). While prior work has assumed access to offline human decisions under support (Laidlaw & Russell, 2021; Charusaie et al., 2022) or oracle queries of human behavior (De-Arteaga et al., 2018; Mozannar & Sontag, 2020) to learn decision support policies, we argue that this data is unrealistic to obtain in practice across all available forms of support *for a new decision-maker*. Thus, for individuals for whom we have no prior information initially, we propose learning how to personalize support *online*. We introduce `THREAD`, an algorithm for learning decision support policies online, to sequentially estimate the prediction error of a decision-maker under each form of support leveraging techniques from the stochastic contextual bandit literature (Li et al., 2010).

Since providing support may be costly (Arbiser et al., 2001; Mimra et al., 2016), it may also be important to consider a second objective regarding the cost of the selected support. We show how `THREAD` can be easily modified to account for a trade-off parameter between performance and cost. However, since it can be challenging to determine an appropriate choice of the trade-off parameter for an unseen decision-maker, we implement a practical hyper-parameter tuning strategy to identify a trade-off parameter using data from a population of decision-makers. This strategy involves finding the set of optimal parameters that minimize the incurred cost of support for each decision-maker while ensuring that a specified threshold is satisfied: we then select a parameter from this set to deploy on a new decision-maker.

We conduct computational experiments to explore the utility of `THREAD` in both the cost-agnostic and cost-aware settings. In the cost-agnostic setting, we characterize the types of decision-maker expertise profiles

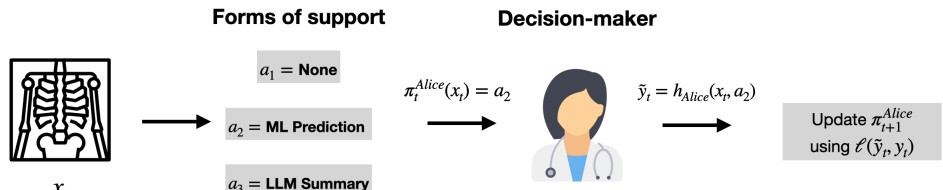

Figure 1: We illustrate the process of learning a decision support policy $\pi_t$ online to improve Alice's performance. For every decision $x_t$, our policy selects a form of support $\pi_t(x_t)$ from a set $\mathcal{A} = \{a_1, a_2, a_3\}$. The policy is updated using $\ell(\tilde{y}_t, y_t)$, the loss incurred after observing Alice's decision $\tilde{y}_t$. Without assuming access to prior offline data, our formulation learns a personalized policy *online*; this means Alice's learned policy may differ from that of another decision-maker, Bob, if they have different expertise. We also consider a cost-aware version of this learning problem, e.g., it is more expensive to elicit an LLM summary.

where THREAD successfully learns policies that outperform offline policies, which are not personalized to the strengths of a new decision-maker, and, demonstrate that if there is no benefit of personalization, THREAD recovers fixed policies, which always provide one form of support. In the cost-aware setting, we show that THREAD can reduce the incurred cost with minimal degradation to performance.

We also conduct human subject experiments ($N = 125$) to test our proposed method on Prolific crowdworkers. We develop Modiste,[1] an interactive tool that gives THREAD an interface. In contrast to prior work that only tests offline policies or evaluates in simulation, we demonstrate how Modiste can be used to learn personalized decision support policies online on both vision and language tasks. We explore forms of support that include expert consensus, outputs from an LLM, or predictions from a classification model. Our findings highlight the importance of running human subject experiments to validate any proposed decision support algorithm and discuss implications of deploying decision support policies in practice. We emphasize our main contributions:

**1. Formalizing decision support policies and their associated algorithms.** We propose a formulation for learning a personalized decision support policy that selects the form of support that maximizes a given decision-maker's performance. We introduce THREAD, an online algorithm to learn such a policy, and provide a multi-objective extension to incorporate the cost of support. We also implement a practical hyper-parameter tuning strategy, which finds a trade-off parameter that minimizes cost and achieves a specified performance threshold. We also begin to characterize under which settings we would expect THREAD to improve performance and/or cost.

**2. Evaluating decision support policies in realistic settings.** We demonstrate the importance of online learning to personalize policies to new decision-makers through both computational and human subject experiments on vision and language tasks. Our human subject experiments, where real users interact with our new tool Modiste, validate our findings from the computational experiments on synthetic decision-makers, demonstrating that THREAD can yield benefits in practice. We plan to release the Modiste interface, THREAD code, and participant data.

## 2 Related Work

**Decision Support.** While various forms of decision support have been proposed, such as expert consensus (Scheife et al., 2015) and changes to machine interfaces (Roda, 2011), more recent forms of support focus on algorithmic tools where decision-makers are aided by machine learning (ML) models (Phillips-Wren, 2012; Bastani et al., 2022; Gao et al., 2021). In some prior work, the human does not always make the final decision, such as those that learn to defer decisions from a model to a single decision-maker (Madras et al., 2018; Mozannar & Sontag, 2020) or others that jointly learn an allocation function between a model and a pool of decision makers (Keswani et al., 2021; Hemmer et al., 2022). In our setting, the *human* is always the decision-maker, even after viewing any form of support: this reflects many real-world applications.

---

[1]While a "modiste" usually refers to someone who uses thread to tailor clothing and make dresses/hats, we use the term to capture our tool's ability to alter a policy to a decision-maker.

This setting includes ones where humans make the final decisions with support from ML models (Green & Chen, 2019; Lai et al., 2023) , as well as ones where humans make decisions when provided with additional information beyond a model prediction (e.g., explanations (Bansal et al., 2021b), uncertainty (Zhang et al., 2020), conformal sets (Babbar et al., 2022)). However, these studies *always* show a single form of support. Our work not only considers the forms of support in these studies but also formalizes *learning* in what contexts each form of support should be provided. An extensive comparison to prior work is in Appendix A.

**Prior Assumptions About Decision-Maker Information.** We briefly survey the assumptions made about the decision-maker when learning decision support policies. The model of the decision maker is either synthetic, thus lacking grounding in actual human behavioral data, or learned from a batch of *offline* annotations (Madras et al., 2018; Okati et al., 2021; Charusaie et al., 2022; Gao et al., 2023). For a new decision-maker or a new form of support, this set of data would not be available in practice. Instead, we propose to learn a decision support policy *online* to circumvent these limitations. Few works use some aspect of online learning for different decision-making settings or under strict theoretical conditions, as we describe in Appendix A.

**Online Learning.** Learning decision support policies can be done with bandit feedback, where at each time step the learner receives a reward based on a selected action. Our set-up is a case of stochastic contextual bandits (Li et al., 2010), where the reward depends on an i.i.d sampled context. In our setting, the contexts are the input, the actions represent the forms of support, and the reward depends on a decision-maker's decision, which in turn provides bandit feedback online. We propose an algorithm to learn decision support policies that can use existing techniques from the contextual bandits literature and evaluate two existing methods, LinUCB (Li et al., 2010) and KNN-UCB (Guan & Jiang, 2018). While we extend the set-up to include a multi-objective problem to incorporate both performance and cost, we cannot directly apply prior works on multi-objective contextual bandits (Tekin & Turğay, 2018; Turgay et al., 2018) because they make theoretical assumptions that are not applicable in our setting, described in Appendix A. Instead, we reduce the multi-objective to a single-objective problem and propose a practical hyper-parameter tuning strategy.

## 3 Formulating and Learning Decision Support Policies

**General Problem Formulation.** We consider a human decision-making process with different forms of decision support. Specifically, we focus on classification tasks where decision-makers can be given a form of support when selecting an outcome (Seger & Peterson, 2013; Lai et al., 2023). In particular, we focus on a multi-class classification problem with observation/feature space $\mathcal{X} \subseteq \mathbb{R}^p$ and outcome/label space $\mathcal{Y} = [K]$. We concentrate on a stochastic setting where the data $(x, y) \in \mathcal{X} \times \mathcal{Y}$ are drawn iid from a fixed, unknown data generating distribution $\mathcal{P}$, an assumption that reflects typical decision-making settings (Bastani & Bayati, 2020; Bastani et al., 2022). In addition, we consider an action set $\mathcal{A}$ corresponding to the forms of support available.[2] Given an observation $x \in \mathcal{X}$, the human attempts to predict the corresponding label $y \in \mathcal{Y}$ using the support prescribed by an action $a \in \mathcal{A}$, i.e., the human makes the prediction using an unknown function $h : \mathcal{X} \times \mathcal{A} \to \mathcal{Y}$. *Our problem formulation and algorithms to solve the formulation are agnostic to the specific forms of support*; we provide multiple instantiations in later human subject experiments. The quality of the final prediction made by the human is measured by a loss function $\ell : \mathcal{Y} \times \mathcal{Y} \to [0, 1]$ w.r.t. the true label. We consider a 0-1 loss function, where $\ell(y, y') = 1$ for $y \neq y'$ and $\ell(y, y') = 0$ for $y = y'$.

**Decision-Making Protocol.** We aim to learn a policy $\pi : \mathcal{X} \to \Delta(\mathcal{A})$ that picks the appropriate form of support to assist a new decision-maker. Let $\Pi$ denote the class of all stochastic decision support policies. Let $\mathcal{A} = \{A_1, \ldots, A_k\}$, and $\pi(x)_{A_i}$ denote $\mathbb{P}[A_i \sim \pi(x)]$ for each $A_i \in \mathcal{A}$. When the policy $\pi$ prescribes the support $A_i$, the human decision-maker makes the prediction $\widetilde{y}$ based on the observation $x$ and support $A_i$, i.e., the final prediction $\widetilde{y}$ is given by $\widetilde{y} = h(x, A_i)$. The human decision-making process with different forms of support is described below. For $t = 1, 2, \ldots, T$:

1. A data point $(x_t, y_t) \in \mathcal{X} \times \mathcal{Y}$ is drawn iid from $\mathcal{P}$.

2. A form of support $a_t \in \mathcal{A}$ is selected using a decision support policy $\pi_t : \mathcal{X} \to \Delta(\mathcal{A})$.

---

[2]A form of support may consist of an individual piece of information (e.g., model prediction) or a particular combination of multiple pieces of information (e.g., model prediction and explanation).

3. The human decision-maker makes the final prediction $\widetilde{y}_t = h(x_t, a_t)$ based on $x_t$ and $a_t$.

4. The human decision-maker incurs a loss $\ell(y_t, \widetilde{y}_t) = 1$ if $y_t \neq \widetilde{y}_t$ and $\ell(y_t, \widetilde{y}_t) = 0$ otherwise.

A static decision support policy would not change throughout the interaction with the decision-maker. This reflects many prior user studies, per Lai et al. (2023); for example, decision-makers may *always* be presented with model predictions. It is possible to update a policy $\pi_t$ iteratively at each time step based on the most recent interaction with the decision-maker, i.e., using $\{(x_t, y_t, a_t, \widetilde{y}_t)\}$. We propose a method to make such an update online in the subsequent section.

**Evaluation of $\pi$ via Expected Loss.** The quality of a policy $\pi$ can be evaluated using the expected loss incurred by the decision-maker across the input space:

$$L_h(\pi) \;=\; \mathbb{E}_{(x,y)\sim\mathcal{P}}\big[\mathbb{E}_{\mathrm{A}_i\sim\pi(x)}[\ell(y, h(x, \mathrm{A}_i))]\big]. \tag{1}$$

We distinguish this metric from the more standard notion of regret, which is typically used to analyze policies in an online learning setting (Li et al., 2010); however, we cannot realize $\pi^*$ for an unseen decision-maker in practical scenarios. Thus, we rely on $L_h(\cdot)$ as a proxy metric for evaluating the effectiveness of $\pi$.

In the cost-agnostic setting, our goal is to find an optimal decision support policy $\pi^*$ that minimizes $L_h(\pi)$. We first rewrite Eq. 1 as follows: $L_h(\pi) = \mathbb{E}_x\big[\sum_{i=1}^k \pi(x)_{\mathrm{A}_i} \cdot r_{\mathrm{A}_i}(x; h)\big]$, where $r_{\mathrm{A}_i}(x; h) = \mathbb{E}_{y|x}[\ell(y, h(x, \mathrm{A}_i))]$ is the human prediction error for input $x$ and support $\mathrm{A}_i$. Then, it can be shown that the optimal policy takes the form $\pi^*(x) = \arg\min_{\mathrm{A}_i \in \mathcal{A}} r_{\mathrm{A}_i}(x; h)$ (see Appendix B.1). This simplified form of $\pi^*$ enables us to propose an efficient strategy to learn the policy online by estimating $r_{\mathrm{A}_i}(x; h)$ values via interaction with the decision-maker $h$. This approach is more tractable than identifying the appropriate modeling assumptions for each $h$ under all forms of support (Kim et al., 2008).

### 3.1 `THREAD`: Learning Personalized Policies

The human decision-making process with various forms of support can be modeled as a stochastic contextual bandit problem, where the forms of support are the arms and $\mathcal{X}$ is the context space. Leveraging this insight, we introduce `THREAD`, outlined in Algorithm 1. `THREAD` is an online algorithm for learning the policy $\pi^*$ that minimizes the expected loss.

---

**Algorithm 1 `THREAD`**

1: **Input:** human decision-maker $h$
2: **Initialization:** data buffer $\mathcal{D}_0 = \{\}$; human error values $\{\widehat{r}_{\mathrm{A}_i,0}(x; h) = 0.5 : x \in \mathcal{X}, \mathrm{A}_i \in \mathcal{A}\}$; initial policy $\pi_1$
3: **for** $t = 1, 2, \ldots, T$ **do**
4:     data point $(x_t, y_t) \in \mathcal{X} \times \mathcal{Y}$ is drawn iid from $\mathcal{P}$
5:     support $a_t \in \mathcal{A}$ is selected using policy $\pi_t$
6:     human makes the prediction $\widetilde{y}_t$ based on $x_t$ and $a_t$
7:     human incurs the loss $\ell(y_t, \widetilde{y}_t)$
8:     update the buffer $\mathcal{D}_t \leftarrow \mathcal{D}_{t-1} \cup \{(x_t, a_t, \ell(y_t, \widetilde{y}_t))\}$
9:     update the decision support policy:

$$\widehat{r}_{\mathrm{A}_i,t}(x; h) \;\leftarrow\; \mathcal{U}_r(\widehat{r}_{\mathrm{A}_i,t-1}(x; h), \mathcal{D}_t), \quad \forall \mathrm{A}_i \in \mathcal{A} \qquad \text{(Step 1)}$$
$$\pi_{t+1}(x) \;\leftarrow\; \mathcal{U}_\pi(\{\widehat{r}_{\mathrm{A}_i,t}\}_i) \qquad \text{(Step 2)}$$

10: **end for**
11: **Output:** policy $\pi_\lambda^{\mathrm{alg}} \leftarrow \pi_{T+1}$

---

$\mathcal{U}_r$**: Update Step 1.** We discuss two approaches to estimate $r_{\mathrm{A}_i}(x; h)$ for all $x \in \mathcal{X}$ and $\mathrm{A}_i \in \mathcal{A}$, but note that any online learning algorithm can be used. We first consider **LinUCB** (Li et al., 2010), a common online learning algorithm that approximates the expected loss $r_{\mathrm{A}_i}(x; h)$ by a linear function $\widehat{r}_{\mathrm{A}_i}(x; h) := \langle \theta_{\mathrm{A}_i}, x \rangle$.

Although the linearity assumption may not hold in general, we learn the parameters $\{\theta_{A_i} : A_i \in \mathcal{A}\}$ using LinUCB with the instantaneous reward function $R(x, y, A_i; h) := -\ell(y, h(x, A_i))$. We then normalize the resulting $\widehat{r}_{A_i}(x; h)$ values to lie in the range $[0, 1]$. The second algorithm we use is an intuitive $K$-nearest neighbor (**KNN**) approach, which is a simplified variant of KNN-UCB (Guan & Jiang, 2018). Here, we maintain an evolving data buffer $\mathcal{D}_t$, which accumulates the history of interactions with the decision-maker. For any new observation $x$, we estimate the $\widehat{r}_{A_i}(x; h)$ values by finding its $K$-nearest neighbors in $\mathcal{D}_t$ and computing the average error of these neighbors.

$\mathcal{U}_\pi$**: Update Step 2.** While one could use a pure exploratory policy $\pi_{t+1}$ in Step 2 of Algorithm 1, where $\pi_{t+1}(x)_{A_i} = 1/|\mathcal{A}|$ for all $A_i \in \mathcal{A}$, this approach would require a large number of interactions $T$ to achieve accurate error estimates. In practical settings, where interactions are limited, as in our human subject experiments, $T$ tends to be relatively small, making this approach less feasible. Thus, we guide exploration with the policy $\pi_{t+1}(x) = \arg\min_{A_i \in \mathcal{A}} \widehat{r}_{A_i,t}(x; h) + b_{A_i,t}(x; h)$ in Step 2, where $b_{A_i,t}(x; h)$ corresponds to some exploration bonus. In Appendix B.3, we provide specific implementations of Algorithm 1 using LinUCB and online KNN.

## 3.2 Extension to Cost-Aware Decision Support

Providing decision support can be expensive (Arbiser et al., 2001; Mimra et al., 2016), so in addition to minimizing loss, it may be desirable to minimize the cost of providing support. We quantify the expected cost of a policy $\pi$ using $c(\pi) = \mathbb{E}_x\big[\sum_{i=1}^{k} \pi(x)_{A_i} \cdot c(A_i)\big]$, where $c(A_i)$ is the cost of providing support $A_i$. Using both objectives, we consider the following multi-objective optimization (MOO) problem: $\min_\pi \mathcal{R}_h(\pi) = [L_h(\pi), c(\pi)]^\top$. We can reformulate this MOO problem into a single-objective optimization (SOO) problem: $\pi_\lambda^* = \arg\min_{\pi \in \Pi} \lambda \cdot L_h(\pi) + (1 - \lambda) \cdot c(\pi)$, where $\lambda \in [0, 1]$. It can be shown that the solutions of the SOO problem can fully characterize the Pareto front of the MOO problem (Mas-Colell et al., 1995; Branke et al., 2008). Formal statements are provided in Appendix B.1. The SOO reformulation allows us to learn the policy $\pi_\lambda^*$ for any $\lambda \in [0, 1]$ using THREAD with minor modifications. Specifically, we can update **Step 2** of Algorithm 1 to $\pi_{t+1}(x) = \arg\min_{A_i \in \mathcal{A}} \lambda \cdot \widehat{r}_{A_i,t}(x; h) + (1 - \lambda) \cdot c(A_i) + b_{A_i,t}(x; h)$ to incorporate the cost of providing support. Since $c(A_i)$ does not depend on the input $x$, it does not affect **Step 1**.

In practice, it may be unclear how to select $\lambda$, but one way to consider this MOO problem may be to learn a decision support policy $\pi$ that achieves a certain level of performance at a minimal cost: given a tolerance threshold $\epsilon \in [0, 1]$ for the expected loss, we can find a policy $\pi$ that achieves the minimum expected cost $c(\pi)$ while maintaining the expected loss $L_h(\pi)$ within $\epsilon$ of the optimal loss $L_h^{\text{opt}}$, where $L_h^{\text{opt}} = \min_\pi L_h(\pi)$. We discuss the existence of such a $\lambda$ for a given $\epsilon$ in Appendix B.1.2.

Given a large number of interactions with a decision-maker $h$, we can simply run Algorithm 1 for multiple values of $\lambda$, i.e., sampling from $\lambda \in [0, 1]$. From the set of corresponding policies learned for each $\lambda$, we can evaluate which $\lambda$ yields a policy with the lowest cost within $\epsilon$ of $L_h^{\text{opt}}$. We test this workflow computationally with synthetic decision-makers in Section 5. However, it is often infeasible to extensively test many choices of $\lambda$ values simultaneously for a new decision-maker without repeating queries of the same input (e.g., as in our human subject experiments); thus, we propose a hyper-parameter tuning strategy using population data. We follow the above workflow to find suitable values for $\lambda$ for each decision-maker in a population. From this set of candidate $\lambda$ values, we select a $\lambda$ to deploy by choosing the most common value (mode) of all $\lambda$ values that yield policies which get decision-makers within $\epsilon$ of their optimal loss. Although the decision-makers in the population may differ from the unseen decision-maker, we prefer this to a random selection of $\lambda$ at deployment. We test our strategy on real users in Section 6.

## 3.3 Expertise Profiles

Decision-makers may have different "expertise" (i.e., strengths and weaknesses) across the input space $\mathcal{X}$ under each form of support. We capture an individual $h$'s expertise via an *expertise profile*, which is defined over the input space $\mathcal{X}$. We divide $\mathcal{X}$ into disjoint regions (i.e., $\mathcal{X} = \cup_{j \in [N]} \mathcal{X}_j$); these regions could be defined by class labels or by covariates, depending on the task.[3] We let $r_{A_i}(\mathcal{X}_j; h)$ denote $h$'s average prediction error under support $A_i$ across region $\mathcal{X}_j$. We define the following three expertise profiles and what we kind of decision support policy we expect to be learned for each:

---

[3]While we instantiate decision-makers this way, THREAD does not assume knowledge of expertise profiles or how they were constructed (e.g., the regions).

- **Approximately Invariant** expertise across all the regions under different forms of support, i.e., $r_{A_1}(\mathcal{X}_j; h) \approx r_{A_2}(\mathcal{X}_j; h) \approx \cdots \approx r_{A_k}(\mathcal{X}_j; h), \forall j \in [N]$. In the cost-aware setting, a decision support policy should learn the fixed form of support that corresponds to picking the cheapest form of support.

- **Varying** expertise where a decision-maker excel in some areas but may benefit from support in areas beyond their training (Schvaneveldt et al., 1985), i.e., $r_{A_1}(\mathcal{X}_j; h) \leq r_{A_2}(\mathcal{X}_j; h)$ and $r_{A_2}(\mathcal{X}_k; h) \leq r_{A_1}(\mathcal{X}_k; h)$, for some $j, k \in [N]$. For this expertise profile, we expect the decision support policy to select different forms of support in different rergions. In particular, we note that the quantity of $|r_{A_1} - r_{A_2}|$ for a region will dictate how efficiently the policy can be learned.

- **Strictly Better** expertise (e.g., $A_1 \succ A_2 \succ \cdots \succ A_k$) that is uniformly maintained across all the regions, i.e., $r_{A_1}(\mathcal{X}_j; h) \leq r_{A_2}(\mathcal{X}_j; h) \leq \cdots \leq r_{A_k}(\mathcal{X}_j; h), \forall j \in [N]$. A decision support policy should learn the fixed form of support to use.

**Instantiating expertise profiles in practice.** To evaluate the utility of `THREAD` under different expertise profiles using *realistic* values for each $r_{A_i}(\mathcal{X}_j; h)$, we collect data on user decisions across different users and then calculate $r_{A_i}(\mathcal{X}_j; h)$. Specifically, we showed each participant similar inputs with different forms of support to estimate $r_{A_i}(\mathcal{X}_j; h)$ for each support $A_i$ in each region $\mathcal{X}_j$. The specific user study we conducted to identify these values is in Section 4.2. The expertise profiles that we collect form a population of decision-makers that we refer to as *human-informed synthetic decision-makers*. From the estimated $r_{A_i}(\mathcal{X}_j; h)$ of each human-informed synthetic decision-maker, we can simulate decision-maker behavior. We leverage these decision-makers to evaluate `THREAD` under multiple values of $\lambda$ in our computational experiments (Section 5) and to select $\lambda$ values to deploy on unseen decision-makers in our human subject experiments (Section 6).

## 4 Experimental Set-up

We overview important details of subsequent experiments. All other details are in Appendix C.

### 4.1 Decision-making Tasks

Our experiments center around the following vision and language datasets:

1. *CIFAR*-10 (Krizhevsky, 2009), a 10-class image classification dataset;

2. *MMLU* (Hendrycks et al., 2020), a multi-task text-based benchmark that tests for knowledge and problem-solving ability across 57 topics spanning both the humanities and STEM.

In terms of the size of $|\mathcal{A}|$, we let $kA$ denote when there are $k$ forms of support for a task. We focus on $k = 2$ or 3, which captures a buffet of real-world scenarios in prior work where decision-makers have a few tools at their disposal, as per Appendix A. The forms of support we study are common in practice (Lai et al., 2023); however, our choices of support are not intended to exhaustively demonstrate the diverse forms of support that our framework can handle. We now describe our two main tasks, which are designed to be accessible to crowdworkers, that will be featured in both the computational and human subject experiments.[4]

**CIFAR-**3*A.* In this task, we consider three forms of support: (1) HUMAN ALONE, where the human makes the decision solely based on the input; (2) MODEL, which shows decision-makers a model's prediction for the given input, (3) CONSENSUS, which shows a distribution over labels from approximately 50 annotators Peterson et al. (2019). Our goal is to construct a setup reflecting a realistic setting in which different forms of support result in different strengths and weaknesses for decision-makers. To instantiate this setting, we deliberately corrupt images of different classes to evoke performance differences – necessitating that a decision-maker appropriately calibrate when to rely on each form of support. We consider 5 of the animal classes in CIFAR-10; of these, we never corrupt images of Birds, do not corrupt images of Deers and Cats for the MODEL, and do

---

[4]In the Appendix, we include a synthetic classification task consisting of well-defined, separable clusters, computational experiments for two additional tasks (Synthetic-2*A* and CIFAR-2*A*), and experiments where we vary the size of $k$.

not corrupt images of Horses and Frogs for the CONSENSUS. We assume the cost of HUMAN ALONE is less than the cost of either form of support, and the cost of MODEL and CONSENSUS is equal.

**MMLU-2$A$.** The two forms of support are HUMAN ALONE and LLM, where the human is provided responses generated from InstructGPT3.5 (`text-davinci-003`) (Ouyang et al., 2022) using the same few-shot prompting scheme for MMLU as Hendrycks et al. (2020). We conducted pilot studies to select a subset of topics where the accuracy of the LLM and average human accuracy vary. We choose the following topics: Computer Science, US Foreign Policy, High School Biology, and Elementary Mathematics. The goal of this task is to evaluate whether we can learn personalized support *"in-the-wild"*, where we naturally expect people to excel at different topics, akin to real-world settings where decision-makers may have "varying" expertise. Again, the cost of MODEL is greater than the cost of HUMAN ALONE.

### 4.2 Human-informed synthetic decision-makers

To construct human-informed synthetic decision-makers with varied and realistic expertise profiles, we recruited 20 participants from Prolific (10 for CIFAR-3$A$ and 10 for MMLU-2$A$). We use the same recruitment scheme as the larger human subject experiment (see Appendix E.1). We define regions of expertise over class labels for CIFAR-3$A$ and over question topics for MMLU-2$A$, as we expect $r_{A_i}(x; h)$ to be roughly constant for $x \in \mathcal{X}_j$ where $\mathcal{X}_j$ is defined by a class label or question topic. On each trial, each participant is randomly assigned a form of support; trials are approximately balanced by the type of support and grouping (i.e., topic or class). We compute participant accuracy averaged over all trials: 100 for CIFAR-3$A$, 60 for MMLU-2$A$. We now describe how we denote expertise profiles in subsequent experiments: assume that there are three regions, an individual's expertise profile under support $A_i$ is written as $r_{A_i} = [0.7, 0.1, 0.7]$, meaning the individual incurs a loss of 0.7 on $\mathcal{X}_1$, 0.1 on $\mathcal{X}_2$, and 0.7 on $\mathcal{X}_3$. In accordance with each task setup, we find participants generally only display varying expertise profiles on CIFAR-3$A$ while we find instances of all three expertise profiles on MMLU-2$A$.

### 4.3 Baselines and Other Parameters

**Algorithms and Baselines.** We compare both variants of `THREAD` (LinUCB/KNN) against the following offline policies:

- *Human and Support*, where the decision-maker *always* receives the same form of support: $\pi(x) = A_i$ for all $x$. In CIFAR-3$A$, there are 3 fixed support baselines, corresponding to each form of support. In MMLU-2$A$, there are 2.

- *Population-level,* where the decision-maker receives a form of support based on the majority vote from 10 learned policies (breaking ties at random). For this baseline, the form of support may vary across contexts but is not personalized to individual needs. We perform the vote based on policies learned for each of the 10 human-informed synthetic decision-makers.

**Number of Interactions.** While more interactions (higher $T$) provide more data points to estimate each $r_{A_i}$, we need to consider what a realistic value of $T$ is given constraints of working with real humans (e.g., limited attention and cognitive load). In traditional online learning, $T$ is usually unreasonably large, on the order of thousands Li et al. (2010); Guan & Jiang (2018). Via pilot studies, we found that 100 CIFAR images or 60 MMLU questions were a reasonable number of decisions to make within 20-40 minutes (a typical amount of time for an online study), which we use throughout our experiments.

## 5 Computational Experiments

We now evaluate `THREAD` on human-informed synthetic decision-makers in both the cost-agnostic (where the goal is to minimize $L_h(\pi) \in (0, 1)$) and cost-aware (where the goal is to minimize both $L_h(\pi)$ and $c(\pi) \in (0, 1)$) settings to characterize `THREAD`'s utility in each setting.

**Cost-agnostic results.** We investigate how `THREAD` compares against offline baselines under each expertise profile (Table 1). We verify that learning decision support policies are not helpful for decision-makers with

Table 1: In the cost-agnostic setting, we evaluate the average excess loss $L_h(\pi) - L_h^{opt}$ (lower is better), and standard deviation across individuals in each expertise profile for both CIFAR-3$A$ (Left)[5]and MMLU-2$A$ (Right). $L_h(\pi)$ is computed by averaging across the last 10 steps of 100 total time steps. We **bold** the variant with the lowest excess loss.

| Algorithm | Invariant | Strictly Better | Varying |
|---|---|---|---|
| H-Only | $0.00 \pm 0.01$ | $0.09 \pm 0.08$ | $0.50 \pm 0.06$ |
| H-Model | $0.00 \pm 0.01$ | $0.22 \pm 0.19$ | $0.35 \pm 0.05$ |
| H-Consensus | $0.00 \pm 0.01$ | $0.23 \pm 0.13$ | $0.27 \pm 0.08$ |
| Population | $0.00 \pm 0.02$ | $0.18 \pm 0.08$ | $0.15 \pm 0.03$ |
| THREAD-LinUCB | $0.00 \pm 0.01$ | $0.17 \pm 0.05$ | $0.19 \pm 0.05$ |
| THREAD-KNN | $0.00 \pm 0.01$ | $\mathbf{0.06 \pm 0.01}$ | $\mathbf{0.08 \pm 0.02}$ |

| Algorithm | Invariant | Strictly Better | Varying |
|---|---|---|---|
| H-Only | $0.01 \pm 0.01$ | $0.18 \pm 0.17$ | $0.22 \pm 0.12$ |
| H-LLM | $0.01 \pm 0.01$ | $0.18 \pm 0.21$ | $0.12 \pm 0.17$ |
| Population | $0.00 \pm 0.02$ | $0.19 \pm 0.07$ | $0.12 \pm 0.09$ |
| THREAD-LinUCB | $0.00 \pm 0.01$ | $0.12 \pm 0.03$ | $0.07 \pm 0.04$ |
| THREAD-KNN | $0.01 \pm 0.01$ | $\mathbf{0.05 \pm 0.03}$ | $\mathbf{0.05 \pm 0.03}$ |

Table 2: In the cost-aware setting for MMLU-2$A$, we compare the expected loss $L_h(\pi)$ and the expected cost $c(\pi)$ (lower is better) averaged across the last 10 steps of 100 total time steps of 5 separate runs. We select two individuals who have a "varying" profile and two individuals who have a "strictly better" profile. We follow the workflow of Section 3.2 to find a suitable $\lambda$, reported for each person in Table 5. We **bold** the algorithm that achieves the lowest cost within $\epsilon = 0.05$ risk of each individual $h$'s $L_h^{\mathrm{opt}}$.

| Algorithm | Varying: $L_h^{\mathrm{opt}} = 0.13$ $r_{\text{H-Only}} = [0.3, 0.1, 0.5, 0.3]$ $r_{\text{H-LLM}} = [0.5, 0.1, 0.0, 0.1]$ | | Varying: $L_h^{\mathrm{opt}} = 0.10$ $r_{\text{H-Only}} = [0.1, 0.3, 0.6, 0.6]$ $r_{\text{H-LLM}} = [0.4, 0.0, 0.3, 0.0]$ | | Strictly Better: $L_h^{\mathrm{opt}} = 0.43$ $r_{\text{H-Only}} = [0.8, 0.6, 0.8, 0.6]$ $r_{\text{H-LLM}} = [0.6, 0.3, 0.7, 0.1]$ | | Strictly Better: $L_h^{\mathrm{opt}} = 0.09$ $r_{\text{H-Only}} = [0.3, 0.9, 0.3, 0.9]$ $r_{\text{H-LLM}} = [0.0, 0.1, 0.0, 0.3]$ | |
|---|---|---|---|---|---|---|---|---|
|  | $L_h(\pi)$ | $c(\pi)$ | $L_h(\pi)$ | $c(\pi)$ | $L_h(\pi)$ | $c(\pi)$ | $L_h(\pi)$ | $c(\pi)$ |
| H-Only | $0.30 \pm 0.03$ | $0.0$ | $0.41 \pm 0.04$ | $0.0$ | $0.68 \pm 0.03$ | $0.0$ | $0.54 \pm 0.05$ | $0.0$ |
| H-LLM | $0.21 \pm 0.03$ | $0.1$ | $0.16 \pm 0.04$ | $0.1$ | $0.42 \pm 0.06$ | $0.1$ | $0.09 \pm 0.02$ | $0.1$ |
| Population | $0.13 \pm 0.03$ | $0.05 \pm 0.01$ | $0.16 \pm 0.04$ | $0.05 \pm 0.01$ | $0.53 \pm 0.05$ | $0.05 \pm 0.01$ | $0.35 \pm 0.07$ | $0.05 \pm 0.01$ |
| THREAD-LinUCB | $0.15 \pm 0.05$ | $0.04 \pm 0.01$ | $\mathbf{0.15 \pm 0.04}$ | $\mathbf{0.06 \pm 0.01}$ | $0.51 \pm 0.06$ | $0.06 \pm 0.01$ | $0.20 \pm 0.10$ | $0.06 \pm 0.01$ |
| THREAD-KNN | $\mathbf{0.17 \pm 0.03}$ | $\mathbf{0.03 \pm 0.01}$ | $0.15 \pm 0.04$ | $0.07 \pm 0.01$ | $\mathbf{0.46 \pm 0.05}$ | $\mathbf{0.05 \pm 0.02}$ | $\mathbf{0.12 \pm 0.03}$ | $\mathbf{0.08 \pm 0.01}$ |

"invariant" expertise profiles. However, for individuals who fall under the "strictly better" and "varying" profiles, we find at least one THREAD variant outperforms offline policies and learns a policy that is closer to the decision-maker's optimal performance. This is because THREAD identifies *which* form of support is better in each context, compared to fixed offline policies which *always* show one form of support or to the population variant, which may not provide the correct form of support to each individual. We note that KNN generally outperforms LinUCB, the latter of which can be saddled by its implicit linearity assumption.

**Cost-aware results.** Using the workflow described in Section 3.2, we aim to learn low-cost policies for each decision-maker that are within $\epsilon = 0.05$ of their optimal loss $L_h^{\mathrm{opt}}$. In these experiments, we focus on MMLU-2$A$, where individuals naturally have "varying" and "strictly better" expertise profiles, two settings it makes sense to consider both loss and cost objectives (Table 2). We omit the "invariant" profile as the best policy is to simply select the cheapest form of support. For individuals who are "strictly better", it may not be possible to further reduce the expected loss compared to one of the offline policies; however, we find that running THREAD can reduce cost while still achieving an expected loss that is within $\epsilon$ of $L_h^{\mathrm{opt}}$. For the individuals with "varying" profiles, we find that THREAD can improve both performance and cost over many of the offline baselines. We find the population baseline is inconsistent in its ability to identify high-performing policies at low cost, in contrast to the personalized policies learned via THREAD. We report similar results for multiple datasets in Appendix D.

**Discussion on parameter selection.** In the cost-aware setting, we used the human-informed synthetic decision-makers to learn policies for multiple values of $\lambda$ simultaneously. However, we observe that often only a subset of the learned policies fall within $\epsilon$ of $L_h^{\mathrm{opt}}$ (e.g., in Figure 8), underscoring that selecting a suitable $\lambda$ may be finicky in practice. We further study the effect of various parameters, e.g., exploration parameters, KNN parameters, embedding size, and the number of interactions in Appendix D. These studies informed the parameter selection of our main text experiments; we advise future work deploying THREAD for new datasets to consider similar experiments with human-informed synthetic decision-makers to understand the impact of these various parameters on decision outcomes.

---

[5]Since CIFAR-2$A$ does not have naturally occurring "invariant" and "strictly better" profiles, we manipulate values identified from the "varying" profile to simulate the other expertise profiles for the sake of completeness in our computational evaluation.

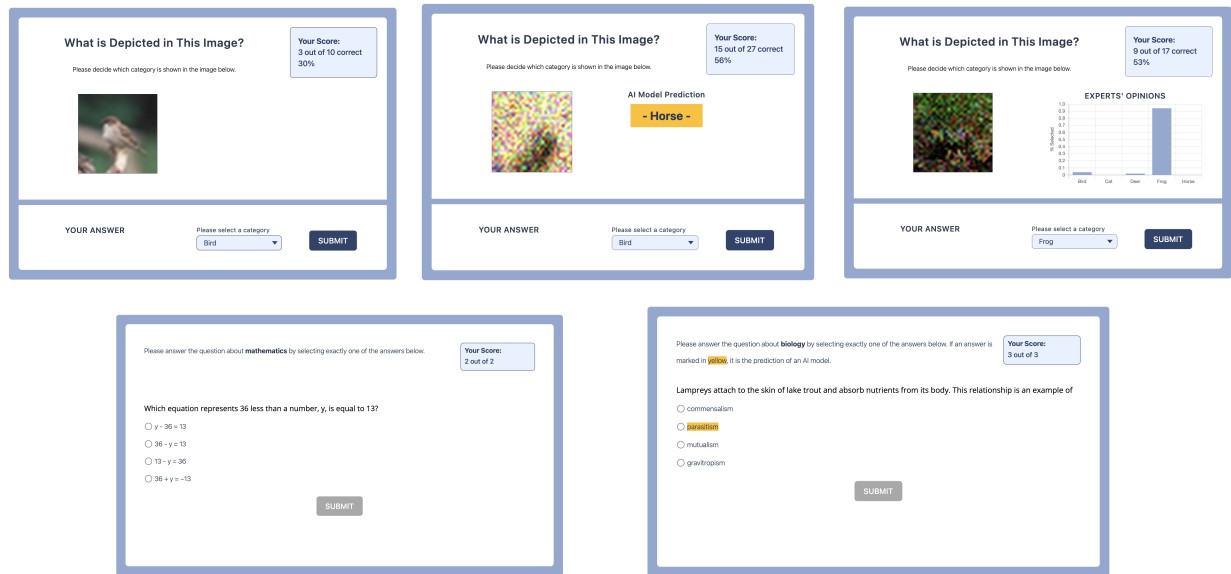

Figure 2: Examples of the `Modiste` interface for each form of support in CIFAR-3*A* (Top) and MMLU-2*A* (Bottom). From left to right, top row: HUMAN ALONE, MODEL, and CONSENSUS; bottom row: HUMAN ALONE and LLM.

# 6 Human Subject Validation

To validate whether decision support policies learned via `THREAD` can improve decision-maker performance and cost in practice, we run a series of human subject experiments, i.e., ethics-reviewed studies with real human participants. We first introduce the set-up of the user study; additional information can be found in Appendix E. To our knowledge, we are one of the first works that run actual user studies to demonstrate the benefits of learning personalized policies online (see our comparison to prior work in Appendix A).

## 6.1 Interactive Interface

To translate our problem formulation into an interactive tool, we create `Modiste`, which provides an interface for `THREAD`. At each time step, `Modiste` sends each user's predictions to a server running `THREAD`, which identifies the next form of support for the next input. `Modiste` then updates the interface accordingly to reflect the selected form of support. Our tool can flexibly be linked with crowdsourcing platforms like Prolific (Palan & Schitter, 2018). In Figure 2, we provide screenshots of the interface under each form of support. Participants are informed of their own correctness after each trial, as well as the correctness of the form of support (e.g., model prediction), if support was provided, so that participants can learn whether support ought to be relied upon.

## 6.2 Recruitment Details

We recruit a total of 125 crowdsourced participants from Prolific to interact with `Modiste` ($N = 45$ and $N = 80$ for CIFAR-3*A* and MMLU-2*A*, respectively). We recruit more participants for MMLU-2*A*, as we expect greater individual differences in regions where support is needed, e.g., some participants may be good at mathematics and struggle in biology, whereas others may excel in biology questions, whereas in CIFAR-3*A*, there is an "optimal" form of support for each stimulus. Each participant is assigned to only one task and one algorithm variant.

**Trade-off parameter selection.** For our cost-agnostic experiments, we let $\lambda = 1.0$ for each variant across all datasets. For our cost-aware experiments, we follow the strategies specified in Section 3.2 to identify $\lambda$ for

Table 3: We report expected loss $L_h(\pi)$ and expected cost $c(\pi)$ incurred (lower is better) in the last 10 trials by Prolific participants for each Algorithm and **bold** the variant with the lowest $L_h(\pi)$. We also consider different choices of $\lambda$, where $\lambda = 1.0$ corresponds to the cost-agnostic setting and $\lambda \neq 1.0$ corresponds to a cost-aware setting, where the choice of $\lambda$ was selected according to Section 3.2.

| Algorithm | $L_h(\pi)$ | $c(\pi)$ |
|---|---|---|
| H-ONLY | $0.68 \pm 0.13$ | $0$ |
| H-MODEL | $0.50 \pm 0.11$ | $0.5$ |
| H-CONSENSUS | $0.32 \pm 0.07$ | $0.5$ |
| Population | $0.66 \pm 0.08$ | $0$ |
| Modiste (LinUCB, $\lambda = 1.0$) | $0.22 \pm 0.15$ | $0.38 \pm 0.06$ |
| Modiste (KNN, $\lambda = 1.0$) | $\mathbf{0.1 \pm 0.06}$ | $\mathbf{0.44 \pm 0.07}$ |
| Modiste (LinUCB, $\lambda = 0.85$) | $0.5 \pm 0.15$ | $0.14 \pm 0.07$ |
| Modiste (KNN, $\lambda = 0.75$) | $0.14 \pm 0.23$ | $0.39 \pm 0.06$ |

| Algorithm | $L_h(\pi)$ | $c(\pi)$ |
|---|---|---|
| H-ONLY | $0.51 \pm 0.14$ | $0$ |
| H-LLM | $\mathbf{0.22 \pm 0.14}$ | $\mathbf{0.1}$ |
| Population | $0.33 \pm 0.19$ | $0.06 \pm 0.01$ |
| Modiste (LinUCB, $\lambda = 1.0$) | $0.25 \pm 0.11$ | $0.05 \pm 0.02$ |
| Modiste (KNN, $\lambda = 1.0$) | $0.3 \pm 0.05$ | $0.07 \pm 0.02$ |
| Modiste (LinUCB, $\lambda = 0.95$) | $0.35 \pm 0.14$ | $0.04 \pm 0.01$ |
| Modiste (KNN, $\lambda = 0.75$) | $0.3 \pm 0.11$ | $0.07 \pm 0.02$ |

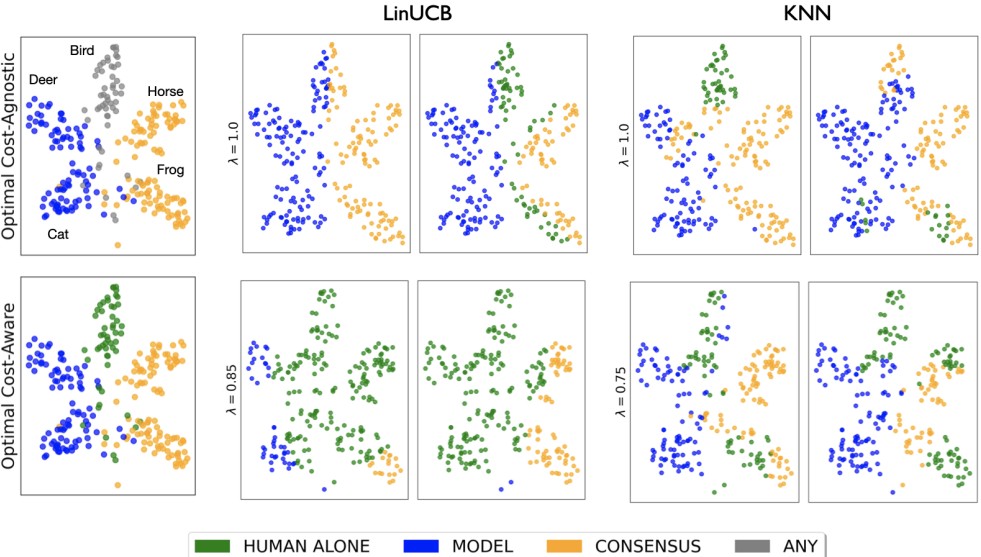

Figure 3: Snapshots of the learned decision support policies computed at the end of 100 interactions with randomly sampled users for CIFAR-3*A*. The forms of support are colored in t-SNE embedding space. In the cost-agnostic setting, both LinUCB and KNN generally identify the optimal form of support for a given input. Since the choice of support for images of Birds does not matter given the task set-up (Section 4.1), we nicely find that when we introduce cost (i.e., when $\lambda \neq 1$), the learned policies favor the cheaper H-ONLY arm for images of Birds. All participant plots are included in Figure 11.

each variant. We visualize this process in further detail in Figure 9. For CIFAR-3*A*, the selected parameter values were $\lambda = 0.85$ for LinUCB and $\lambda = 0.75$ for KNN. For MMLU-2*A*, the selected parameter values were $\lambda = 0.95$ for LinUCB and $\lambda = 0.75$ for KNN.

## 6.3 Results

**CIFAR-3*A*.** By design, the task compels "varying" profiles: `Modiste`'s forte. We find that `Modiste` learns to reconstruct near-optimal policies in both the standard and cost-aware settings, as depicted in Figure 3. This is reflected quantitatively in Table 3, where both `Modiste` variants, particularly KNN, have lower expected losses than any of the offline policies. We also identify a choice of $\lambda$ for each algorithm that modulates loss versus cost, by using our human-informed synthetic decision-makers. In particular, this is done by using the cheaper form of support (H-ONLY) more. However, the trade-off parameter affects method differently, as the gap between the KNN variants is much smaller than that between the LinUCB variants.

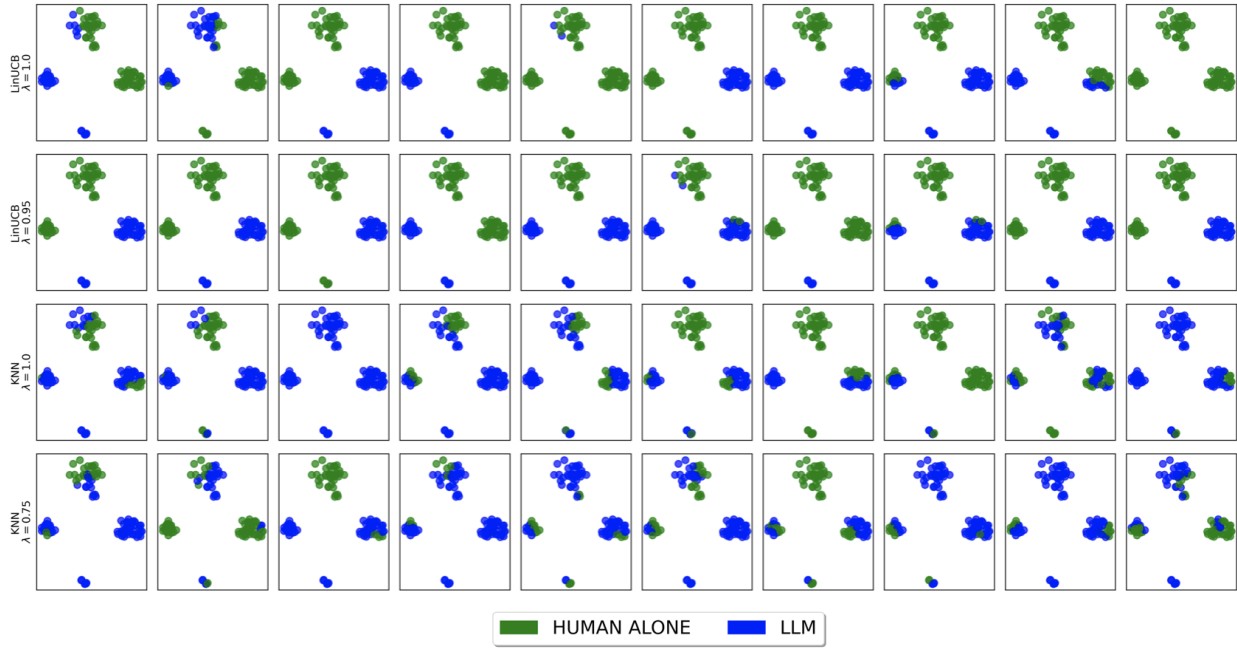

Figure 4: Snapshots of the learned decision support policies computed at the end of 60 interactions with randomly sampled users for MMLU-2*A*. The forms of support are colored in t-SNE embedding space. Topics correspond to clusters, clockwise from the top: math, biology, computer science, foreign policy.

**MMLU-2*A*.** Polymaths are rare; we observe no different in our Prolific data, as most participants struggle with at least one topic. Many individuals benefited from receiving the LLM as support, due in part to our task design (wherein the LLM excels at three of the four topics). In Figure 4, we see that `Modiste` personalizes support to decision-maker expertise, yielding policies that provide support on different topics for different decision-makers. In Table 3, `Modiste` is not statistically different from the best baselines (H-LLM) in terms of expected loss. Yet, we observe lower expected costs for both LinUCB and KNN in the cost-agnostic setting, and find similar results, as expected, in the cost-aware setting. This suggests that learning policies with `Modiste` has merit: participants who have no expertise can expect to receive support from the LLM, and those who are proficient in one topic, but not in another, can expect to see the LLM in only cases beyond their expertise. While $L_h(\pi)$ of the `Modiste` variants are similar to that of the Population baseline in Table 3, the variance is significantly smaller – particularly with KNN – underscoring the benefits of personalization.

### 6.4 Discussion

Our human subject experiments validate trends observed in the computational experiments in both the cost-agnostic and cost-aware settings, demonstrating that `Modiste` can be used to learn decision support policies online. We find that `THREAD` enjoys performance and cost benefits when encountering users with "varying" expertise profiles, and recovers fixed policies for participants with "strictly better" profiles in the standard setting. While our human-informed synthetic decision-makers permit us to pick a suitable $\lambda$ for participants, future work modeling human idiosyncrasies (Steyvers & Kumar, 2022) explicitly in synthetic decision-makers may bolster `THREAD`'s effectiveness. Participants' reliance is one such factor. We define a metric for "reliance sensibility," as the proportion of trials the participant agreed with the support when correct and responding differently when the support was incorrect, out of all trials where the participant received support. The higher the proportion, the more "sensible" a participant's degree of reliance on the provided support is, relative to what would be most beneficial for decision outcomes[6]. We plot the performance

---

[6]Note that we cannot directly deduce whether, or to what degree, an individual participant relied on the form of support when presented - since we do not have access to what an individual would have said without support; alternative reliance inference schemes like (Tejeda et al., 2022) could be considered in future work.

of the learned policy (i.e., incurred loss) against a participants' reliance sensibility in Figures 5, respectively. While in CIFAR-3$A$, participants largely had calibrated reliance, we find a negative correlation between expected loss and reliance (Pearson $r$ correlation $= -0.47$ and $-0.59$ for KNN and LinUCB, respectively) in MMLU-2$A$. These data underscore the importance of understanding when decision-makers may over-rely on potentially fallible support (Buçinca et al., 2021; Bussone et al., 2015).

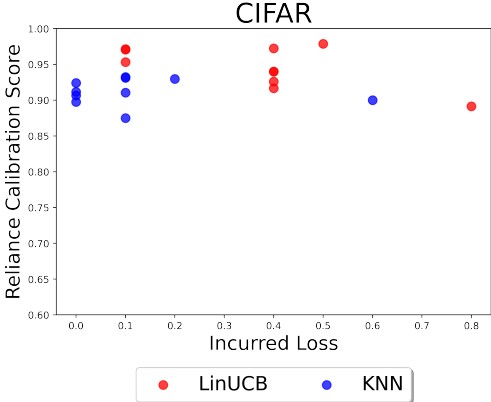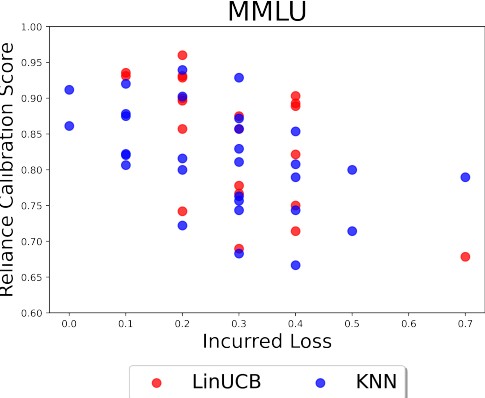

Figure 5: Relationship between a participants' sensibility of reliance (measured as the proportion of times they correctly agreed or disagreed with the form of support's prediction) and the loss incurred by the learned policy for the participant. Reliance is computed over all trials (100 for CIFAR-3$A$, 60 for MMLU-2$A$), and loss is averaged over the final 10 timesteps.

**Limitations.** While we obtain promising results with `Modiste`, we acknowledge that we only consider the classification setting here where we get immediate feedback (e.g., we can calculate loss to update $\pi$). We imagine that extending to a delayed feedback setting or to a different cognitive task (e.g., planning or perception) would prove fruitful. As with most online learning algorithms, selecting a suitable explore-exploit trade-off (a la Table 6) can prove challenging, though we find that our learned policy is preferable to the fixed policies used in practice at the moment. Though the use of `THREAD` is promising, to encourage the thoughtful deployment of personalized decision support policies, we urge consideration of the following. First, we note that significant issues can arise when decision-makers blindly rely on decision support (Buçinca et al., 2020), especially when the support is erroneous or ineffective; such over-reliance requires careful attention to prevent. Second, our problem definition hinges on domain experts defining the available forms of support; i.e., we need a clearly defined $\mathcal{A}$ to use `Modiste`. In practice, this may prove difficult, as one may not know how to define specific forms of support or decision-makers may have access to varying support sets due to regulatory or organizational reasons.

## 7    Conclusion

A decision support policy captures when and which form of support should be provided to improve a decision-maker's performance. To the best of our knowledge, we are the first to consider learning such a policy online for unseen decision-makers. We propose `THREAD`, an algorithm for learning a decision support policy online, and then extend it to handle auxiliary objectives, like the cost of support, using a tuned trade-off parameter. We instantiate two variants of `THREAD` using stochastic contextual bandits. We perform computational and human subject experiments to highlight the importance—and feasibility—of personalizing decision support policies for individual decision-makers. Our human subject experiments are encouraging, demonstrating that we can learn decision support policies in remarkably few iterations and tease apart differences in decision-makers' need for support. While encouraging rich cross-talk between domain experts and ML practitioners, future work exploring the integration of `Modiste` into existing decision-making workflows would pave a route towards responsible deployment of decision support.

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

We provide further details on decision support policies, additional computational experiments with `THREAD`, and extensive information on our human subject experiments with `Modiste`.

## A Comparison Against Prior Work

Most papers on human-AI collaboration have considered clever ways of abstaining from prediction on specific inputs (Cortes et al., 2016; 2018), learning deferral functions based on multiple experts (Vovk, 1998; Keswani et al., 2021), or teaching decision-makers when to rely (Mozannar et al., 2022b). There are also a number of papers from the HCI literature (see survey by (Lai et al., 2023)) that evaluate the two-action setting of our formulation using a *static* policy (e.g., always showing the ML model prediction or always showing some form of explanation).

To clarify how our set-up and assumptions differ from prior work, we overview work that we believe could be considered most similar to ours. We decompose our comparisons along a few dimensions: **Decision-support set-up:** Does the human make the final decision, or is it a different set-up? **Assumptions about decision-maker information:** What does prior work assume about access to a decision-maker when learning a policy? **Evaluation:** Does prior work simulate humans? Does prior work run user studies?

Mozannar & Sontag (2020):

- **Decision-support set-up:** This work's set-up can be considered a two-action setting of our formulation, where $\mathcal{A} = \{\texttt{DEFER}, \texttt{MODEL}\}$. Extending the work of Madras et al. (2018), this work proposes the learning to defer paradigm, where the decision-maker may not always make a final decision (i.e., sometimes the decision is deferred entirely to an algorithmic-based system). In our set-up, deferring to a $\texttt{MODEL}$ is equivalent to always adhering to a label-based form of support. The human is always the final decision-maker in our work, which is representative of many decision-making set-ups in practice (Lai et al., 2022), but not captured in this line of prior work.

- **Assumptions about decision-maker information:** This work assumes oracle query access to the decision-maker, for whom they are learning a policy.

- **Evaluation:** This work evaluates their approach using human simulations (no real human user studies). Mozannar & Sontag (2020) define synthetic experts in the following way: "if the image belongs to the first $k$ classes the expert predicts perfectly, otherwise the expert predicts uniformly over all classes."

Gao et al. (2021) and Gao et al. (2023):

- **Decision-support set-up:** This work defines two actions: $\mathcal{A} = \{\texttt{DEFER}, \texttt{MODEL}\}$. They do not consider the, more practical assumption that the decision-maker will view a model prediction before making a decision themselves. Their formulation is similar to the above but they use offline bandits to learn a suitable policy.

- **Assumptions about decision-maker information:** Gao et al. (2021) assume that understanding decision-maker's expertise (at a population-level, not at a individual-level) can help learn better routing functions (i.e., defer only when appropriate). Gao et al. (2023) assume access to a decision history for each decision-maker.

- **Evaluation:** They run a human subject experiment to collect offline annotations, which can be used to learn when to defer to decision-makers. Gao et al. (2023) goes further to personalize a deferral policy based on offline annotations for each decision-maker.

Bordt & Von Luxburg (2022):

- **Decision-support set-up:** This work's set-up can be considered a two-action setting of our formulation, where $\mathcal{A} = \{\texttt{DEFER}, \texttt{SHOW}\}$; however, they are concerned with the learnability of such a set up. They do not devise algorithms for this setting, as they are only focused on its theoretical formulation.

- **Assumptions about decision-maker information:** They assume the decision-maker has access to information not contained in the input but still important to the task.

- **Evaluation:** This is a theory paper, containing neither computational nor human subject experiments.

Noti & Chen (2022):

- **Decision-support set-up:** This work's set-up can be considered as a two-action setting of our formulation, where $\mathcal{A} = \{\texttt{DEFER}, \texttt{SHOW}\}$. This is not an online algorithm and as such, the policy does not update.

- **Assumptions about decision-maker information:** They assume access to a dataset of human decisions and that all decision-makers are similar (i.e., they deploy one policy for all decision-makers).

- **Evaluation:** This work is one of few that runs a user study to evaluate their (fixed) policy on unseen decision-makers.

Babbar et al. (2022):

- **Decision-support set-up:** This work considers the two-action setting of our formulation, where $\mathcal{A} = \{\texttt{DEFER}, \texttt{CONFORMAL}\}$. Their policy is learned offline, is the same for all decision-makers, and is not updated in real-time based on decision-maker behavior.

- **Assumptions about decision-maker information:** They use CIFAR-10H Peterson et al. (2019) to learn a population-level deferral policy. This assumes that we have annotations for each decision-maker for every datapoint and assumes that all new decision-makers have the same expertise profiles as the population average.

- **Evaluation:** This work runs a user study to evaluate their (fixed) policy. They show the benefits of `DEFER+CONFORMAL` over `CONFORMAL` or `SHOW` alone.

Wolczynski et al. (2022):

- **Decision-support set-up:** This work considers two actions per our formulation, where $\mathcal{A} = \{\texttt{DEFER}, \texttt{SHOW}\}$. They learn a rule-based policy offline for each decision-maker.

- **Assumptions about decision-maker information:** They simulate human behavior by considering explicit functions of how human expertise may vary in input space.

- **Evaluation:** While this work does consider the human to be the final decision-maker, they only validate their proposal in simulation, not on actual human subjects.

We now list various forms of support that can be included in the action space of our problem formulation. The design of the action space is up to domain experts, who can decide not only which actions are feasible but also how much cost to assign to each form of support.

- `DEFER`: This form of support is equivalent to **no support**. Decision-makers are asked to make a decision without any assistance. The machine learning community has studied how to identify when to defer to a subset of examples to humans based on human strengths Bansal et al. (2021a); Wilder et al. (2021) and/or model failures Chow (1957); Geifman & El-Yaniv (2017). The premise of such an action would be to allow human decision-makers to be unaided and squarely placing decision liability on the individual.

- `SHOW`: In many settings, machine learning (ML) models are trained to do prediction tasks similar to the decision-making task prescribed to the human, or in the case of foundation models (Bommasani et al., 2021), ML systems can be adapted to aid decision-making, even if the task was not specifically prescribed at train-time (Yang et al., 2023b). In essence, a machine learning model prediction, or associated generation (e.g., a code snippet (Mozannar et al., 2022a)) would be shown to aid an individual decision-maker. This has been shown to help improve decision-maker performance. The following are variations of showing a model prediction to a decision-maker.

- **CONFORMAL**: For classification tasks, only displaying the most likely label may not lead to good performance due to various reasons, including uncertainty in the modeling procedure (Vovk et al., 2005; Bondi et al., 2022); however, such uncertainty can be communicated to decision-maker by showing a prediction set to experts (Babbar et al., 2022). Such a prediction set might be generated using conformal prediction, which guarantees the true label lies in the set with a user-specified error tolerance (Bates et al., 2021; Straitouri et al., 2022).

- **CONFIDENCE**: Instead of translating the uncertainty into a prediction set (or interval), one could simply show the confidence or uncertainty of the prediction, which may manifest as displaying probabilities, standard errors, or entropies Spiegelhalter (2017); Bhatt et al. (2021). The visualization mechanism used for displaying confidence may alter the decision-maker's performance (Hullman et al., 2018; Zhang et al., 2020).

- **EXPLAIN**: In addition to providing a model prediction, many have considered showing an explanation of model behavior, examples of which include feature attribution (Ribeiro et al., 2016; Buçinca et al., 2020), sample importance (Kim et al., 2014; Jeyakumar et al., 2020), counterfactual explanations Ustun et al. (2019); Antoran et al. (2020), and natural language rationales Ehsan et al. (2018); Camburu et al. (2018). Displaying such explanations to end users has had mixed results on how decision-making performance is affected Chen et al. (2022); Lai et al. (2022). Worryingly, in many settings, showing some types of explanations may lead to to over-reliance on models by giving the perception of competence Buçinca et al. (2020); Zerilli et al. (2022); Chen et al. (2023).

- **CONSENSUS**: One can also depict forms of support that are independent of any model, for instance, presenting the belief of one or more humans. Belief distributions can be constructed by pooling over many different humans' "votes" for what a label ought to be, e.g., Peterson et al. (2019); Beyer et al. (2020); Uma et al. (2022; 2020); Gordon et al. (2021; 2022), or by eliciting distributions over the likely label directly from each individual human Collins et al. (2022; 2023). These consensus distributions permit the expression of uncertainty *without any model*. However, the elicitation of this form of support may be costly and humans may be fallible in the information they provide, e.g., due to direct labeling errors (Dawid & Skene, 1979; Augustin et al., 2017; Whitehill et al., 2009; Wei et al., 2022), or miscalibrated confidence O'Hagan et al. (2006); Collins et al. (2023); Lichtenstein et al. (1977); Tversky & Kahneman (1996).

- **ADDITIONAL**: While much of this paper focused on support that provides decision-makers with label information, decision support may also entail acquiring or displaying additional contextual information (e.g., new features Bakker et al. (2021)) or requesting previously unseen features, for instance, through additional medical diagnostics Harrell et al. (1982); Mylonakis et al. (2000). This flavor of support can be varied structurally, ranging from the results of a search query (Nakano et al., 2021) to hierarchical information like exposing the subsidiary ownership structure for multinational corporations Erramilli (1996). In terms of the cost, some pieces of additional information may require additional cost, or certification if pertaining to sensitive attributes.

**Prior work on multi-objective contextual bandits.** We summarize why some theoretical work on multi-objective contextual bandits cannot be directly applied to our problem formulation. In their study, Tekin & Turğay (2018) addressed a contextual multi-armed bandit problem with two objectives, where one objective dominates the other. Their aim was to maximize the total reward in the non-dominant objective while ensuring that the dominant objective's total reward is also maximized. However, our specific case requires the minimization of the total expected cost of the support policy while ensuring that the expected accuracy of the decision-maker under the policy remains above a certain threshold. Therefore, the techniques presented by here cannot be directly applied to our scenario. Turgay et al. (2018) investigated the multi-objective contextual bandit problem with similarity information. Their approach relies on the assumption that a Lipschitz condition holds for the set of feasible context-arm pairs concerning the expected rewards for all objectives and that the learning algorithm has knowledge of the corresponding distance function (see Assumption 1 of their paper).

## B    Additional Details on Problem Formulation

### B.1    Details on Problem Formulation

**Standard (cost-agnostic) Setting.** The optimization problem in the standard setting, where the only objective relates to expected loss, can be written as follows:

$$
\begin{aligned}
\min_{\pi\in\Pi} L_h(\pi) \;=\; & \min_{\pi\in\Pi}\mathbb{E}_{(x,y)\sim\mathcal{P}}\big[\mathbb{E}_{A_i\sim\pi(x)}[\ell(y,h(x,A_i))]\big] \\[4pt]
& \overset{(a)}{=} \min_{\pi\in\Pi}\mathbb{E}_{(x,y)\sim\mathcal{P}}\bigg[\sum_{i=1}^{k}\pi(x)_{A_i}\cdot\ell(y,h(x,A_i))\bigg] \\[4pt]
& \overset{(b)}{=} \min_{\pi\in\Pi}\mathbb{E}_x\bigg[\sum_{i=1}^{k}\pi(x)_{A_i}\cdot\mathbb{E}_{y|x}[\ell(y,h(x,A_i))]\bigg] \\[4pt]
& \overset{(c)}{=} \min_{\pi\in\Pi}\mathbb{E}_x\bigg[\sum_{i=1}^{k}\pi(x)_{A_i}\cdot r_{A_i}(x;h)\bigg],
\end{aligned}
$$

where $(a)$ is due to the notation $\pi(x)_{A_i} := \mathbb{P}[A_i\sim\pi(x)]$, $(b)$ is due to the properties of expectation, and $(c)$ is due to the notation $r_{A_i}(x;h) := \mathbb{E}_{y|x}[\ell(y,h(x,A_i))]$. Then, by noting the fact that the expression $\mathbb{E}_x\big[\sum_{i=1}^{k}\pi(x)_{A_i}\cdot r_{A_i}(x;h)\big]$ can be optimized independently for each $x\in\mathcal{X}$, we can rewrite the above optimization problem as follows, for each $x\in\mathcal{X}$:

$$
\min_{\pi(x)\in\Delta(\mathcal{A})}\sum_{i=1}^{k}\pi(x)_{A_i}\cdot r_{A_i}(x;h) \;=\; \min_{A_i\in\mathcal{A}} r_{A_i}(x;h).
$$

Thus, an optimal policy for the above optimization problem is: $\pi^*(x)\in\arg\min_{A_i\in\mathcal{A}} r_{A_i}(x;h)$ with random tie-breaking.

**Cost-Aware Setting.** In the cost-aware setting, we consider the following multi-objective optimization (MOO) problem:

$$
\min_{\pi}\mathcal{R}_h(\pi) = [L_h(\pi), c(\pi)]^\top, \tag{2}
$$

where $L_h(\pi) = \mathbb{E}_{(x,y)\sim\mathcal{P}}\big[\mathbb{E}_{A_i\sim\pi(x)}[\ell(y,h(x,A_i))]\big]$ is the expected cost of the policy $\pi$, and $c(\pi) = \mathbb{E}_x\big[\mathbb{E}_{A_i\sim\pi(x)}[c(A_i)]\big]$ is the expected cost of the policy $\pi$. We can reformulate this MOO problem into a single-objective optimization (SOO) problem as follows:

$$
\min_{\pi\in\Pi}\lambda\cdot L_h(\pi) + (1-\lambda)\cdot c(\pi), \tag{3}
$$

where $\lambda\in[0,1]$. It can be shown that the solutions of the SOO problem can fully characterize the Pareto front of the MOO problem (Mas-Colell et al., 1995; Branke et al., 2008). For any $\lambda\in[0,1]$, we can rewrite the SOO problem as follows:

$$
\begin{aligned}
\min_{\pi\in\Pi}\lambda\cdot L_h(\pi) + (1-\lambda)\cdot c(\pi) \;=\; & \min_{\pi\in\Pi}\lambda\cdot\mathbb{E}_{(x,y)}\big[\mathbb{E}_{A_i\sim\pi(x)}[\ell(y,h(x,A_i))]\big] + (1-\lambda)\cdot\mathbb{E}_x\big[\mathbb{E}_{A_i\sim\pi(x)}[c(A_i)]\big] \\[4pt]
& \overset{(a)}{=} \min_{\pi\in\Pi}\lambda\cdot\mathbb{E}_{(x,y)}\bigg[\sum_{i=1}^{k}\pi(x)_{A_i}\cdot\ell(y,h(x,A_i))\bigg] + (1-\lambda)\cdot\mathbb{E}_x\bigg[\sum_{i=1}^{k}\pi(x)_{A_i}\cdot c(A_i)\bigg] \\[4pt]
& \overset{(b)}{=} \min_{\pi\in\Pi}\lambda\cdot\mathbb{E}_x\bigg[\sum_{i=1}^{k}\pi(x)_{A_i}\cdot\mathbb{E}_{y|x}[\ell(y,h(x,A_i))]\bigg] + (1-\lambda)\cdot\mathbb{E}_x\bigg[\sum_{i=1}^{k}\pi(x)_{A_i}\cdot c(A_i)\bigg] \\[4pt]
& \overset{(c)}{=} \min_{\pi\in\Pi}\mathbb{E}_x\bigg[\sum_{i=1}^{k}\pi(x)_{A_i}\cdot[\lambda\cdot r_{A_i}(x;h) + (1-\lambda)\cdot c(A_i)]\bigg],
\end{aligned}
$$

where $(a)$ is due to the notation $\pi(x)_{A_i} := \mathbb{P}[A_i \sim \pi(x)]$, $(b)$ is due to the properties of expectation, and $(c)$ is due to the notation $r_{A_i}(x; h) := \mathbb{E}_{y|x}[\ell(y, h(x, A_i))]$. Then, by noting the fact that the expression $\mathbb{E}_x\left[\sum_{i=1}^{k} \pi(x)_{A_i} \cdot [\lambda \cdot r_{A_i}(x; h) + (1-\lambda) \cdot c(A_i)]\right]$ can be optimized independently for each $x \in \mathcal{X}$, we can rewrite the above optimization problem as follows, for each $x \in \mathcal{X}$:

$$\min_{\pi(x) \in \Delta(\mathcal{A})} \sum_{i=1}^{k} \pi(x)_{A_i} \cdot [\lambda \cdot r_{A_i}(x; h) + (1-\lambda) \cdot c(A_i)] = \min_{A_i \in \mathcal{A}} \lambda \cdot r_{A_i}(x; h) + (1-\lambda) \cdot c(A_i).$$

Thus, an optimal policy for the SOO optimization problem is:

$$\pi_\lambda^*(x) \in \underset{A_i \in \mathcal{A}}{\arg\min} \, \lambda \cdot r_{A_i}(x; h) + (1-\lambda) \cdot c(A_i), \tag{4}$$

with random tie-breaking.

### B.1.1 Pareto Optimality

We use dominance (Miettinen, 2008) to define Pareto optimality for the MOO problem in Eq. 2.

**Definition B.1** (Dominant policy)**.** A policy $\pi$ is said to dominate another policy $\pi'$, noted as $\pi \prec \pi'$, if the following holds: (i) $L_h(\pi) \leq L_h(\pi')$ and $c(\pi) \leq c(\pi')$; and (ii) either $L_h(\pi) < L_h(\pi')$ or $c(\pi) < c(\pi')$. Likewise, we denote $\pi \preceq \pi'$ if $\pi' \nprec \pi$.

**Definition B.2** (Pareto front and Pareto optimality)**.** Given a set of policies $\Pi$, the set of Pareto front policies is $\mathcal{P}_\Pi = \{\pi \in \Pi : \nexists \pi' \in \Pi \text{ s.t. } \pi' \prec \pi\}$. The corresponding Pareto front is given by $\mathcal{P}_\Pi^{\mathcal{R}_h} = \{(v_1, v_2) \in \mathbb{R}^2 : \exists \pi \in \mathcal{P}_\Pi \text{ s.t. } v_1 = L_h(\pi) \text{ and } v_2 = c(\pi)\}$. A policy $\pi$ is a Pareto optimal solution to the MOO problem in Eq. 2 iff $\pi \in \mathcal{P}_\Pi$.

**Definition B.3** (Convex Pareto front)**.** A Pareto front $\mathcal{P}_\Pi^{\mathcal{R}_h}$ is convex if $\forall v, v' \in \mathcal{P}_\Pi^{\mathcal{R}_h}, \lambda \in [0, 1], \exists v_\lambda \in \mathcal{P}_\Pi^{\mathcal{R}_h}$ such that $v_\lambda \preceq \lambda \cdot v + (1-\lambda) \cdot v'$.

Note that optimal solutions to the SOO problem in Eq. 3 are Pareto optimal solutions to the MOO problem in Eq. 2. In the following proposition, we show that the Pareto front $\mathcal{P}_\Pi^{\mathcal{R}_h}$ of the MOO problem in Eq. 2 can be fully characterized by the solutions of the SOO problem in Eq. 3, i.e., any Pareto optimal policy $\pi \in \mathcal{P}_\Pi$ is a solution to the SOO problem in Eq. 3 for some choice of $\lambda \in [0, 1]$.

**Proposition B.4.** *The Pareto front of the MOO problem in Eq. 2 is convex:*
$\forall v, v' \in \mathcal{P}_\Pi^{\mathcal{R}_h}, \lambda \in [0, 1], \exists v'' \in \mathcal{P}_\Pi^{\mathcal{R}_h} : v'' \preceq \lambda \cdot v + (1-\lambda) \cdot v'$. *Every Pareto solution of the MOO problem in Eq. 2 is a solution to the SOO problem in Eq. 3:* $\forall v \in \mathcal{P}_\Pi^{\mathcal{R}_h}, \exists \lambda : v = [L_h(\pi_\lambda^*), c(\pi_\lambda^*)]^\top$.

*Proof.* First, we note that the set of stochastic policies $\Pi$ is a convex set, and both $L_h : \Pi \to [0, 1]$ and $c : \Pi \to [0, 1]$ are convex functions. Then, the first statement of the proposition follows from Theorem 4.1 of (Martinez et al., 2020). The second statement is a direct application of the results in (Geoffrion, 1968). $\square$

### B.1.2 Learning Policy with a Tolerance Threshold

In practice, given the set of Pareto optimal policies $\Pi_h^{\text{opt}} := \{\pi_\lambda^* : \lambda \in [0, 1]\}$ corresponding to the MOO problem in Eq. 2, it can be challenging to determine which policy to select from this set. One possible strategy for making this choice is to opt for a policy that attains a certain level of performance at the lowest possible cost. To formalize this, we introduce a tolerance threshold $\epsilon \in [0, 1]$ for the expected loss. We then seek a decision support policy $\pi$ that minimizes the expected cost while keeping the expected loss within $\epsilon$ of the optimal loss $L_h^{\text{opt}} = \min_\pi L_h(\pi)$, i.e., we consider the following constrained optimization problem:

$$\pi^\epsilon \in \underset{\pi \in \Pi}{\arg\min} \, c(\pi) \quad \text{such that} \quad L_h(\pi) \leq L_h^{\text{opt}} + \epsilon. \tag{5}$$

An illustration of the Pareto front for the MOO problem in Eq. 2 can be found in Figure 6. This visualization highlights that the aforementioned policy $\pi^\epsilon$ belongs to the set of Pareto optimal policies $\Pi_h^{\text{opt}}$, i.e., there exists a $\lambda(\epsilon) \in [0, 1]$ such that $\pi_{\lambda(\epsilon)}^* = \pi^\epsilon$.

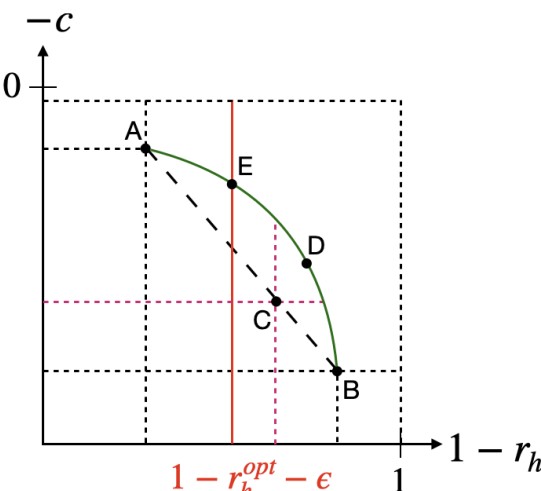

Figure 6: The green curve corresponds to the (unknown) Pareto front of the MOO problem in Eq. 2, e.g., $\mathsf{A} = (1 - L_h(\pi_0^*), -c(\pi_0^*))$, $\mathsf{B} = (1 - L_h(\pi_1^*), -c(\pi_1^*))$, and $\mathsf{D} = (1 - L_h(\pi_\lambda^*), -c(\pi_\lambda^*))$ for some $\lambda \in (0,1)$. The point $\mathsf{E} = (1 - L_h(\pi^\epsilon), -c(\pi^\epsilon))$ corresponds to the constrained optimisation problem Eq. 5, which is the point on the curve that we would like our decision support policy to achieve. The dashed line corresponds to the set $\{\pi_\mu := \mu \cdot \pi_0^* + (1-\mu) \cdot \pi_1^* : \mu \in [0,1]\}$, e.g., $\mathsf{C} = (1 - L_h(\pi_\mu), -c(\pi_\mu))$ for some $\mu \in (0,1)$. While the set of Pareto optimal policies $\Pi_h^{\mathrm{opt}}$ may be difficult to compute without further assumptions, the set $\{\pi_\mu\}$ can be recovered if one is able to compute $\pi_0^*$ and $\pi_1^*$, which would already require accurate estimation of $L_h(x,a)$ values for all $x$ and $a$. Note that the Pareto front is convex according to Proposition B.4.

## B.2  Human Misspecification

We upper bound the misspecification error of applying a policy $\pi_h$ (originally learned for human $h$) on human $\widetilde{h}$. Inspired by (Ben-David et al., 2006), we define $\Pi\Delta\Pi$-divergence to bound the human misspecification error. Let $\ell_h(\pi, \pi') = \mathbb{E}_x[\ell(h(x, \pi(x)), h(x, \pi'(x)))]$ be the expected disagreement between two policies $\pi$ and $\pi'$ for human $h$. The $\Pi\Delta\Pi$-divergence between two humans $h$ and $\widetilde{h}$ is defined as $d_{\Pi\Delta\Pi}(h, \widetilde{h}) = \sup_{\pi, \pi' \in \Pi} \left| \ell_h(\pi, \pi') - \ell_{\widetilde{h}}(\pi, \pi') \right|$. Based on this divergence measure, we bound the risk $L_{\widetilde{h}}(\pi_h) = \mathbb{E}_{(x,y) \sim \mathcal{P}}[\ell(y, \widetilde{h}(x, \pi_h(x)))]$ of using policy $\pi_h$ on human $\widetilde{h}$ as follows:

$$L_{\widetilde{h}}(\pi_h) \leq L_h(\pi_h) + d_{\Pi\Delta\Pi}(h, \widetilde{h}) + \lambda_\Pi,$$

where $\lambda_\Pi = \inf_{\pi' \in \Pi}[L_h(\pi') + L_{\widetilde{h}}(\pi')]$. The result mentioned above is derived by directly adapting the proof of Theorem 1 presented in (Ben-David et al., 2006).

## B.3  Implementations of THREAD

The human decision-making process with various forms of support can be effectively modeled as a stochastic contextual bandit problem. In this model, the diverse forms of support represent the available arms, and $\mathcal{X}$ represents the context space. For our investigation, we leverage two simple and efficient techniques from the existing contextual bandit literature: LinUCB (Li et al., 2010) and KNN-UCB (Guan & Jiang, 2018). It's worth noting that while any contextual bandit algorithm could be employed, we chose these two simpler methods as they have demonstrated their effectiveness in empirically showcasing the utility of our online decision support policy learning framework.

We present specific implementations of the THREAD algorithm (see Algorithm 1) using LinUCB and KNN-UCB in Algorithms 2 and 3, respectively. Importantly, these algorithms naturally inherit the regret bounds associated with LinUCB and KNN-UCB, as discussed below.

In the case of Algorithm 2, under the linear realizability assumption $\lambda \cdot \mathbb{E}_{y|x}[\ell(y, h(x, a))] + (1-\lambda) \cdot c(a) = x^\top \theta_a^*$ for all $x \in \mathcal{X} \subseteq \mathbb{R}^d$ and $a \in \mathcal{A}$, and the constraints $\|x\|_2 \leq 1$ and $\|\theta_a^*\|_2 \leq 1$, we have (Li et al., 2010; Abbasi-Yadkori et al., 2011):

$$\text{REGRET}(T) \;=\; \mathbb{E}\left[\sum_{t=1}^{T} \lambda \cdot \ell(y_t, h(x_t, a_t)) + (1-\lambda) \cdot c(a_t) - \lambda \cdot \ell(y_t, h(x_t, a_t^*)) - (1-\lambda) \cdot c(a_t^*)\right] \;=\; \widetilde{\mathcal{O}}(dK\sqrt{T}),$$

where $a_t^* = \arg\min_{a \in \mathcal{A}} x_t^\top \theta_a^*$, $d$ is the dimension of $x_t$, and $K$ is the number of available support forms.

For Algorithm 3, given the Lipschitz condition $|\lambda \cdot \mathbb{E}_{y|x}[\ell(y, h(x, a))] + (1-\lambda) \cdot c(a) - \lambda \cdot \mathbb{E}_{y|x'}[\ell(y, h(x', a))] - (1-\lambda) \cdot c(a)| \leq L \cdot \|x - x'\|_2$ for all $x, x' \in \mathcal{X}$ and $a \in \mathcal{A}$, we have (Guan & Jiang, 2018):

$$\text{REGRET}(T) \;=\; \mathbb{E}\left[\sum_{t=1}^{T} \lambda \cdot \ell(y_t, h(x_t, a_t)) + (1-\lambda) \cdot c(a_t) - \lambda \cdot \ell(y_t, h(x_t, a_t^*)) - (1-\lambda) \cdot c(a_t^*)\right] \;=\; \widetilde{\mathcal{O}}\left(\frac{1+d}{2+d} KT^{\frac{1+d}{2+d}}\right),$$

where where $a_t^* = \arg\min_{a \in \mathcal{A}} \lambda \cdot \mathbb{E}_{y|x_t}[\ell(y, h(x_t, a))] + (1-\lambda) \cdot c(a)$.

---

**Algorithm 2** `THREAD` using LinUCB

---

1: **Input:** trade-off parameter $\lambda$; human decision-maker $h$; cost function $c : \mathcal{A} \to [0, 1]$; UCB parameter $\alpha$
2: **Initialization:** data buffer $\mathcal{D}_0 = \{\}$; $\mathbf{A}_a = \mathbf{I}_{p \times p}$ and $\mathbf{b}_a = \mathbf{0}_{p \times 1}$ for all $a \in \mathcal{A}$; $\{\theta_a = (\mathbf{A}_a)^{-1}\mathbf{b}_a : a \in \mathcal{A}\}$; human prediction error values $\{\widehat{r}_{a,0}(x; h) = \langle \theta_a, x \rangle : x \in \mathcal{X}, a \in \mathcal{A}\}$; initial policy $\pi_1(x)_a = 1/|\mathcal{A}|$ for all $a \in \mathcal{A}$
3: **for** $t = 1, 2, \ldots, T$ **do**
4:     data point $(x_t, y_t) \in \mathcal{X} \times \mathcal{Y}$ is drawn iid from $\mathcal{P}$ (normalized s.t. $\|x_t\|_2 \leq 1$)
5:     support $a_t \in \mathcal{A}$ is selected using policy $\pi_t$
6:     human makes the prediction $\widetilde{y}_t$ based on $x_t$ and $a_t$
7:     human incurs the loss $\ell(y_t, \widetilde{y}_t)$
8:     update the buffer $\mathcal{D}_t \leftarrow \mathcal{D}_{t-1} \cup \{(x_t, a_t, \ell(y_t, \widetilde{y}_t))\}$
9:     update the decision support policy:

$$\begin{aligned}
\mathbf{A}_{a_t} &\leftarrow \mathbf{A}_{a_t} + x_t x_t^\top \\
\mathbf{b}_{a_t} &\leftarrow \mathbf{b}_{a_t} + r_t x_t \\
\theta_a &\leftarrow (\mathbf{A}_a)^{-1}\mathbf{b}_a \text{ for all } a \in \mathcal{A} \\
\widehat{r}_{a,t}(x; h) &\leftarrow \langle \theta_a, x \rangle \text{ for all } a \in \mathcal{A} \quad\quad\quad\quad (\text{Step 1}) \\
\pi_{t+1}(x) &\leftarrow \arg\min_{a \in \mathcal{A}} \lambda \cdot \widehat{r}_{a,t}(x; h) + (1-\lambda) \cdot c(a) - \alpha \cdot \sqrt{x^\top (\mathbf{A}_a)^{-1} x} \quad (\text{Step 2})
\end{aligned}$$

10: **end for**
11: **Output:** policy $\pi_\lambda^{\text{alg}} \leftarrow \pi_{T+1}$

---

### B.4 $\lambda$ Selection Strategies

There are many ways to select the $\lambda$ to use on unseen decision-makers using a population of decision-makers. We outline three such strategies below. Note we use B in the main paper.

A. **Most likely $\lambda$.** For each simulator, we identify the set of $\{\lambda_{\text{sim-}j}\}$, which yield policies that meet $L_{h_{\text{sim-}j}}^{\text{opt}} + \epsilon$. We then take the $\lambda$ that occurs the *most often* across simulators. If no policy meets the threshold for $h_{\text{sim-}j}$, we select the policy closest to $L_{h_{\text{sim-}j}}^{\text{opt}}$, which can be computed given $r_{A_i}$ for all $x$ for each simulated decision-maker.

B. **Most likely $\lambda$ with lowest cost.** For each simulator, we identify the set of $\{\lambda_{\text{sim-}j}\}$, which yield policies that meet $L_{h_{\text{sim-}j}}^{\text{opt}} + \epsilon$, and select the $\lambda$ whose policy has the least expected cost. Then, we identify the most common value across simulators. While the set of values here is a subset of the

---

**Algorithm 3** `THREAD` using online KNN

---

1: **Input:** trade-off parameter $\lambda$; human decision-maker $h$; cost function $c : \mathcal{A} \to [0,1]$; warm-up steps $W$; number of neighbours $K$; exploration parameter $\gamma$

2: **Initialization:** data buffer $\mathcal{D}_0 = \{\}$; human prediction error values $\{\widehat{r}_{a,0}(x;h) = 0.5 : x \in \mathcal{X}, a \in \mathcal{A}\}$; initial policy $\pi_1(x)_a = 1/|\mathcal{A}|$ for all $a \in \mathcal{A}$

3: **for** $t = 1, 2, \ldots, T$ **do**

4:     data point $(x_t, y_t) \in \mathcal{X} \times \mathcal{Y}$ is drawn iid from $\mathcal{P}$

5:     support $a_t \in \mathcal{A}$ is selected using policy $\pi_t$

6:     human makes the prediction $\widetilde{y}_t$ based on $x_t$ and $a_t$

7:     human incurs the loss $\ell(y_t, \widetilde{y}_t)$

8:     update the buffer $\mathcal{D}_t \leftarrow \mathcal{D}_{t-1} \cup \{(x_t, a_t, \ell(y_t, \widetilde{y}_t))\}$

9:     update the decision support policy:

$$\mathcal{N}(x) \leftarrow \text{K neighbouring data points for } x \text{ in } \mathcal{D}_t$$

$$\mathcal{N}_a(x) \leftarrow \{(x_i, a_i, \ell(y_i, \widetilde{y}_i)) : (x_i, a_i, \ell(y_i, \widetilde{y}_i)) \in \mathcal{N}(x) \text{ and } a_i = a\} \quad \text{for all } a \in \mathcal{A}$$

$$\widehat{r}_{a,t}(x;h) \leftarrow \frac{1}{|\mathcal{N}_a(x)|} \cdot \sum_{(x_i, a_i, \ell(y_i, \widetilde{y}_i)) \in \mathcal{N}_a(x)} \ell(y_i, \widetilde{y}_i) \quad \text{for all } a \in \mathcal{A} \text{ with } |\mathcal{N}_a(x)| > 0 \quad \text{(Step 1)}$$

$$\pi_{\text{rand}}(x)_a \leftarrow \frac{1}{|\mathcal{A}|} \quad \text{for all } a \in \mathcal{A}$$

$$\pi_{\text{knn}}(x) \leftarrow \underset{a \in \mathcal{A}}{\arg\min} \; \lambda \cdot \widehat{r}_{a,t}(x;h) + (1 - \lambda) \cdot c(a)$$

$$\pi_{t+1}(x) \leftarrow \pi_{\text{rand}}(x) \quad \text{if } t \leq W \quad \text{(Step 2)}$$

$$\pi_{t+1}(x) \leftarrow \gamma \cdot \pi_{\text{rand}}(x) + (1 - \gamma) \cdot \pi_{\text{knn}}(x) \quad \text{if } t > W \quad \text{(Step 2)}$$

10: **end for**

11: **Output:** policy $\pi_\lambda^{\text{alg}} \leftarrow \pi_{T+1}$

---

selection strategy A, a cost-aware selection strategy may identify a parameter that leads to a lower cost for new decision-makers.

C. **Conservative $\lambda$.** For each simulator, we identify $\lambda_{\text{sim-}j}$ which is the minimum parameter that meets $L_{h_{\text{sim-}j}}^{\text{opt}} + \epsilon$. We then take $\max_j \lambda_{\text{sim-}j}$ over all simulated decision-makers. In the absence of suitable population-level information, this strategy is similar to picking a conservative value of $\lambda$ that prioritizes performance.

# C    Additional Experimental Details

## C.1    Datasets

### C.1.1    Computational Only

Two tasks (Synthetic-2$A$ and CIFAR-2$A$) are used only in the computational experiments, so we describe them abstractly. Both have the same set-up, but have different underlying data generating distributions. Consider learning a decision support policy in a setting where there are *two* forms of support available: $\mathcal{A} = \{A_1, A_2\}$, hence CIFAR-2$A$. Let $c(A_1) = 0$ and $c(A_2) = 0.5$. We simulate a decision-maker's behavior under each $r_{A_i}$ across 3 classes for Synthetic/CIFAR. We instantiate the $m$ decision-makers in the population data using human simulations, where we randomly sample $r_{A_i}(x; h)$ from a distribution for each $h$ in the population.

### C.1.2    CIFAR

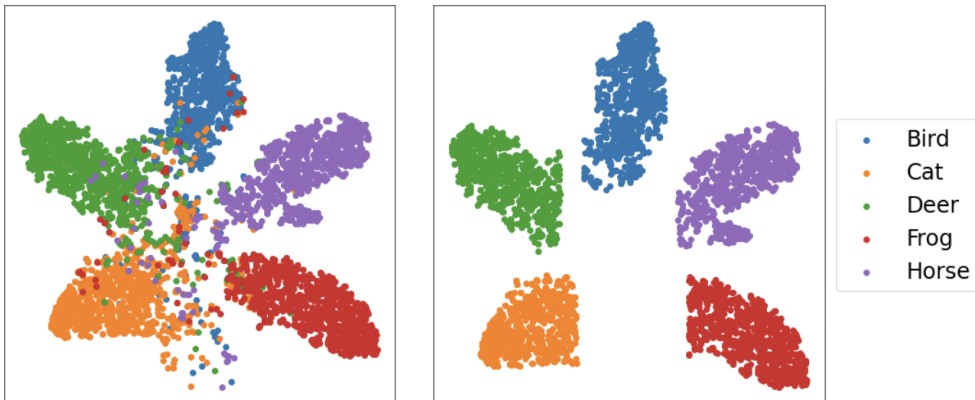

Figure 7: We depict the latent space of the CIFAR subset used. Embeddings without filtering (left) and after filtering out points that violate separability of classes (right). We use the embeddings on the right for our Synthetic-2$A$ task and the left for the natural CIFAR tasks.

To explore the performance of our algorithms in both separable and non-separable settings, we construct two subvariants of the CIFAR-10 (Krizhevsky, 2009) image dataset. As depicted in Figure 7, we subsample from the CIFAR embeddings to create linearly separable classes. Embeddings are constructed by running t-SNE on the 512-dimensional latent codes extracted by the penultimate layer of a variant of the VGG architecture (VGG-11) (He et al., 2016) trained on the animal class subset of CIFAR; the model attained an accuracy of 89.5%. We filter the original 10 CIFAR classes to only include those involving animals (Birds, Cats, Deers, Dogs, Frogs, and Horses). We consider at most 5 such classes in our experiments (dropping Dog).

### C.1.3    MMLU

We consider questions from four topics of MMLU (elementary mathematics, high school biology, high school computer science, and US foreign policy). Topic names match those proposed in (Hendrycks et al., 2020). We select topics to cover span an array of disciplines across the sciences and humanities, as discussed in Appendix E. We use questions in the MMLU test set for each topic that are at most 150 characters long. We limit the length of questions to facilitate readability in our human user studies and wanted to maximize parity between our computational and human experiments by considering the same set of questions across both set-ups. This yields 264, 197, 98, and 47 questions for the elementary mathematics, high school biology, US foreign policy, and high school computer science topics, respectively. We emphasize that in CIFAR, our forms of support operate in label space – in contrast, with MMLU, support selection is in *covariate* (topic) space – further highlighting the flexibility of our adaptive support paradigm.

We leverage OpenAI's LLM-based embedding model (`text-embedding-ada-003`), to produce embeddings over the question prompts. Embedding vectors extracted via OpenAI's API are by default length 1536; we apply t-SNE, like in the CIFAR tasks, to compress the embeddings into two-dimensional latent codes per example. Nicely, these latents are already separable (see Figure 4; no post-filtering is applied to ensure separability).

## C.2 CIFAR Task Set-up

We consider a 5-class subset of CIFAR (Bird, Cat, Deer, Frog, and Horse), as discussed in Appendix D[7]. We instantiate two forms of support: 1) a simulated AI model which provides a prediction for the image class, and 2) a consensus response derived from real humans, derived from the approximately 50 human annotators from CIFAR-10H (Peterson et al., 2019; Battleday et al., 2020), presented as a distribution over the 5 classes.

We treat the original CIFAR-10 test set labels as the "true" labels and only include images for which the original CIFAR-10 label matches the label deemed most likely from the CIFAR-10H annotators (discrepancy was a rare occurrence, only 1.1% of the 5000 images considered). We sample a pool of 300 such images, and construct three different batches of 100 images. Participants are assigned to one batch of 100 images. Images per batch are sampled in a class-balanced fashion (i.e., 20 images for each of the 5 categories). Images are shuffled for each participant. The same latent codes ($z_t$) used in the computational experiments are employed for the user studies. For CIFAR, we define the support costs as: $c(\text{HUMAN ALONE}) = 0$ and $c(\text{MODEL}) = c(\text{CONSENSUS}) = 0.5$.

### C.2.1 Corruptions and Support Design

We want to study whether our algorithms can properly learn which forms of decision support are actually needed by real humans. We therefore need to ensure that the task is sufficiently rich such that humans do *need* support (and the existing forms of support which can be of value). To mimic such a setting in CIFAR, we deliberately corrupt images of all classes except one (i.e., Birds[8]) when presenting images to the user. Corruptions are formed via a composition of natural adversarial transformations proposed in Hendrycks et al. (2020), specifically shot noise ontop of glass blur. This enables us to check that our algorithms are able to properly recommend support for all other classes, as a human will be unable to decide on image category unassisted. Example corrupted images can be seen in Figure 2.

To ensure that the available forms of support have different regions of strength, we enforce that each form of support is only good at two of the five classes. In the case of the simulated AI model, we return the "true" class whenever the image is a Deer or Cat classes, and return one of the incorrect classes for all other images. For the consensus labels, we use the CIFAR-10H distribution derived from 50 annotators when the image is a Horse or Frog. As the CIFAR-10H labels were originally collected over all 10 CIFAR classes, sometimes an annotation was endorsed for a class outside of the 5 we consider; in that case, we discard the annotations assigned to those classes and renormalize the remaining distribution (on average, $< 2\%$ of the original labeling mass was discarded per image). For all other classes, we intend the consensus to be an unhelpful form of support, as such, for images in the Bird, Deer, and Horse class, we return a uniform distribution (i.e., providing no information to the participant).

## C.3 MMLU Task Set-up

Participants respond to 60 multiple choice questions, which were balanced by topic (15 questions per topic). Participants are assigned to one of three possible batches of 60 such questions. Order is shuffled. Participants are informed of the topic associated with each question (e.g., that the question was about biology). We implement a 10 second delay between when the question is presented and when the participant is allowed

---

[7]We explored a 3 class variant for HSEs to directly match the computational experiments; however, we realized that participants were able to figure out that which classes were impoverished, raising the base rate of correctly categorizing such images. Such behavior again highlights the need to carefully consider real human behavior in adaptive decision support systems.

[8]We further disambiguate the non-corrupted bird class by upsampling to 160x160 images via Lanczos-upsampling following (Peterson et al., 2019; Battleday et al., 2020; Collins et al., 2022) before presenting them to participants.

to submit their response to encourage participants to try each problem in earnest. For MMLU, we let the support costs be $c(\text{HUMAN ALONE}) = 0$ and $c(\text{MODEL}) = 0.1$.

### C.3.1  Topic Selection

We ran several pilot studies with `Modiste` to determine people's base performance on a subset of MMLU topics. We intended to select a diverse set of topics such that it was unlikely any one participant would excel at all topics – as many of the topics were specialized and challenging (Hendrycks et al., 2020) – but also varied enough that participants may be strong in at least one area. We found that a large number of participants achieved reasonable performance on elementary mathematics, and that a sufficient number of participants also excelled at questions in the high school biology, US foreign policy, and high school computer science topics – though usually not all together.

We also factored in `InstructGPT-3.5`'s performance while deciding on the topics. To best check whether our adaptive decision support algorithms are effective at learning good policies – like with CIFAR – it is helpful to have support available that is effective, should someone be unable to answer adequately alone. As a result, we looked more sympathetically on categories where model accuracy was already high.

However, the real-world is not so perfect: there may not be high-performing support available when a human struggles to make a decision. As such, we also deliberately forced down the accuracy of the LLM form of support in the mathematics topic. While we expect that most participants would be able to solve elementary mathematics problems without the aid of the LLM support, should they be unable to, the LLM was not able to help them. We leave further impovershing studies which mimic real-world support settings for future work.

### C.3.2  Question Selection and Model Accuracy

The resulting accuracy of the LLM form of support on the questions shown to humans per topic is 29% for elementary mathematics, 89% for high school biology, 87% for US foreign policy, and 91% for high school computer science. We upweighted selection of foreign policy questions that participants in our pilot had gotten correct when constructing the question subset, as we wanted to ensure that, should a participant have foreign policy experience, they would be able to answer them.

### C.4  Compute Resources

All computational experiments were run using CPUs (either on local machines or Google Colab), with the exception of training the VGG used for CIFAR embeddings. Here, we accessed a group compute cluster and trained the model for roughly five hours on a Nvidia A100-SXM-80GB GPU.

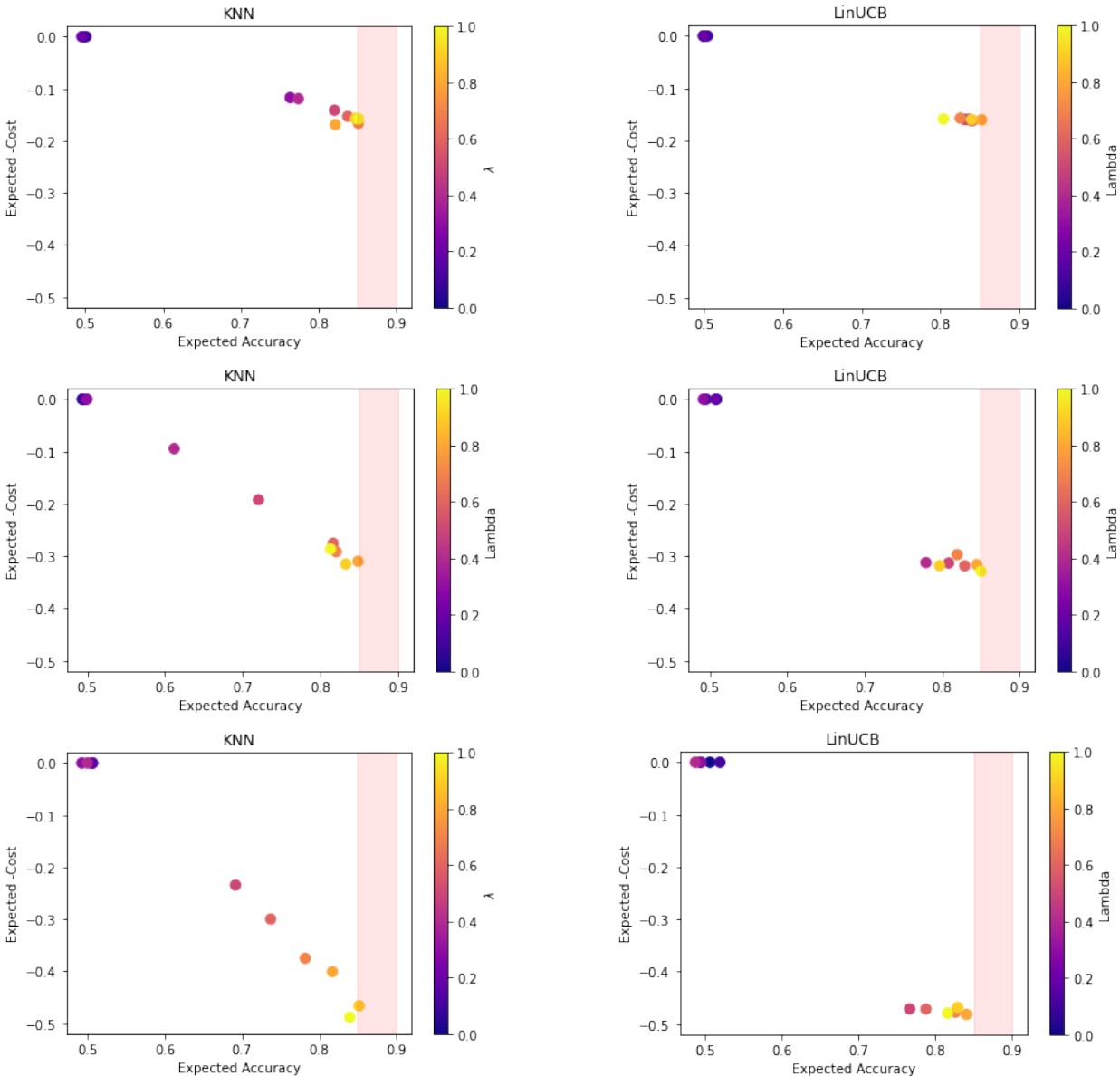

Figure 8: We plot the expected accuracy and the negative of the expected cost for an individual sampled from the Synthetic-2*A* dataset for $\lambda \in [0, 1]$. The ideal policy would lie in the far right corner. The red region denotes policies that fall within $\epsilon$ of the best risk for the sampled individual. We observe that there exists a policy that lies in the red region for both THREAD-KNN and THREAD-LinUCB. We also vary the cost structure (i.e., the cost of $A_1 = 0$ and the cost of $A_2 = 0.25, 0.5, 0.75$ in the first, second, and third row respectively).

## D    Additional Computational Experiments

**Implications of selecting the trade-off parameter $\lambda$ (with varying cost structure).**    We follow the strategies specified in Section 3.2 to identify $\lambda$ for our cost-aware experiments. In Figure 9, we visualized the expected excess loss and cost for each value of $\lambda$ that is used for our human-informed synthetic decision-makers. Further, as shown in Figure 8, we highlight how only a subset of $\lambda$ values falls within $\epsilon = 0.05$ of the $L_h^{\text{opt}}$ of that individual (e.g., in Figure 8). We also find that THREAD-KNN has more monotonic and controllable behavior as $\lambda$ increases, while $\lambda$ seems to have less of an effect on THREAD-LinUCB. We note that the set of $\lambda$

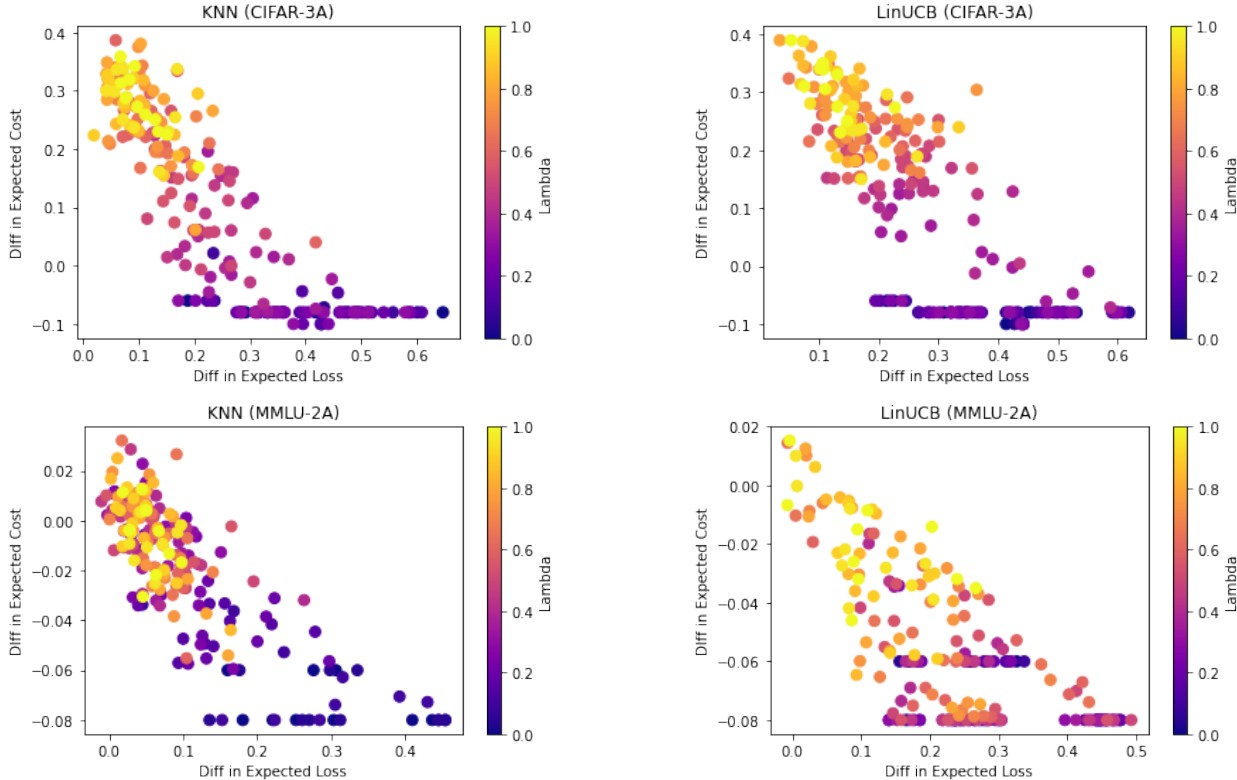

Figure 9: Top is CIFAR-3*A* and Bottom is MMLU-2*A*. For each of the $N = 10$ simulators, we instantiate using the population data, we compute the difference in expected loss between the optimal policy for that simulator and the learned policy for all values of $\lambda \in [0, 1]$ in increments of 0.05, and compute the difference in expected cost in the same manner. For KNN, we observe overlap between many values of $\lambda$ that all come within $\epsilon$ of the Best Risk: this implies we can use lower values of $\lambda$ to potentially achieve the desired level of performance. For LinUCB, we find that large values of $\lambda$ are required to achieve a suitable level of performance.

values that fall within the red region is affected by the choice of cost structure, as shown in Figure 8. Thus, any changes to the cost structure would imply the need to redo the hyper-parameter tuning process.

**Additional Experiments on Datasets.** In Table 4, we provide additional computational results for CIFAR-2*A* and MMLU-2*A* in the cost-aware setting. We find that `THREAD` successfully personalizes decision support policies that trade-off cost and performance effectively. We note that in all of our experiments, we fix the exploration parameters in LinUCB and KNN (i.e., $\alpha = 1$ and $\gamma = 0.1$ respectively). However, we do explore varying these exploration parameters in Table 6.

**Varying KNN Parameters.** While LinUCB does not require identifying additional parameters, KNN has two: $K$ which is the number of nearest neighbors to select when estimating the risk of a form of support and $W$ which is the length of the warm-up period. In Table 7, we show that as long as $K$ is reasonably sized (i.e. $K > 3$), the performance of KNN does not vary too much across datasets.

**Varying Embedding Size.** In the main text, we run computational experiments using a two-dimension t-SNE embedding. In Table 8 for CIFAR-2*A*, we consider how `THREAD` would behave when we vary dimensionality of $\mathcal{X}$. We extract higher dimensional embeddings from animal-class trained VGGs (see above) with varied penultimate layer widths, where width matches embedding size. While we opt for t-SNE embeddings, in the main paper, to provide a clearer visualizations, we find that strong performance holds even for high-dimensional embeddings. We expect that even larger embedding dimensions will permit `THREAD` to work in

Table 4: Unlike Table 2 where risks come from pilot studies, we instantiate simulated **synthetic** humans, each specified by the human's loss on each form of support H-ONLY (Human only) and H-MODEL (Human+Model), for MMLU-2$A$. We report the expected loss $r_h(\pi)$ and the expected cost $c(\pi)$—for both metrics, lower is better—averaged across the last 10 steps of 100 total time steps along with their standard deviations (across 5 runs). We bold the algorithm that achieves the lowest cost within $\epsilon$ of the Best Risk for each human simulation and find that our algorithm outperforms baselines on various simulated humans. For each form of support, we specify risk $r$ over each of the 4 topics for MMLU-2$A$. We fix the cost structure: $c(\text{H-ONLY}) = 0.0$ and $c(\text{H-MODEL}) = 0.1$. The selected $\lambda$ values used for THREAD are in Table 5. For THREAD, we select the best policy over a sweep of $\lambda$ values that minimizes Eq. 5 under $\epsilon = 0.05$. We also indicate the values of $\lambda$ chosen beneath the results table.

| | Algorithm | $r_{A_1} = [0.7, 0.1, 0.7]$ $r_{A_2} = [0.1, 0.1, 0.1]$ | | $r_{A_1} = [0.7, 0.1, 0.7]$ $r_{A_2} = [0.1, 0.7, 0.1]$ | | $r_{A_1} = [0.7, 0.1, 0.7]$ $r_{A_2} = [0.7, 0.7, 0.1]$ | |
| --- | --- | --- | --- | --- | --- | --- | --- |
| | | $L_h(\pi)$ | $c(\pi)$ | $L_h(\pi)$ | $c(\pi)$ | $L_h(\pi)$ | $c(\pi)$ |
| | $A_1$ Only | $0.51 \pm 0.05$ | $0.0$ | $0.50 \pm 0.06$ | $0.0$ | $0.50 \pm 0.06$ | $0.0$ |
| | $A_2$ Only | $0.10 \pm 0.03$ | $0.5$ | $0.30 \pm 0.05$ | $0.5$ | $0.50 \pm 0.04$ | $0.5$ |
| | Population | $0.37 \pm 0.04$ | $0.13 \pm 0.03$ | $0.34 \pm 0.05$ | $0.13 \pm 0.02$ | $0.49 \pm 0.06$ | $0.12 \pm 0.03$ |
| Synthetic-2$A$ | THREAD-LinUCB | $\mathbf{0.15 \pm 0.05}$ | $\mathbf{0.31 \pm 0.03}$ | $0.12 \pm 0.04$ | $0.32 \pm 0.03$ | $0.34 \pm 0.11$ | $0.28 \pm 0.07$ |
| | THREAD-KNN | $0.11 \pm 0.04$ | $0.36 \pm 0.03$ | $\mathbf{0.13 \pm 0.04}$ | $\mathbf{0.30 \pm 0.03}$ | $\mathbf{0.33 \pm 0.06}$ | $\mathbf{0.21 \pm 0.03}$ |
| | Best Risk | $0.1$ | $0.33$ | $0.1$ | $0.33$ | $0.3$ | $0.17$ |

| | Algorithm | $r_{A_1} = [0.7, 0.1, 0.7]$ $r_{A_2} = [0.1, 0.1, 0.1]$ | | $r_{A_1} = [0.7, 0.1, 0.7]$ $r_{A_2} = [0.1, 0.7, 0.1]$ | | $r_{A_1} = [0.7, 0.1, 0.7]$ $r_{A_2} = [0.7, 0.7, 0.1]$ | |
| --- | --- | --- | --- | --- | --- | --- | --- |
| | | $L_h(\pi)$ | $c(\pi)$ | $L_h(\pi)$ | $c(\pi)$ | $L_h(\pi)$ | $c(\pi)$ |
| | $A_1$ Only | $0.50 \pm 0.05$ | $0.0$ | $0.50 \pm 0.05$ | $0.0$ | $0.47 \pm 0.05$ | $0.0$ |
| | $A_2$ Only | $0.1 \pm 0.03$ | $0.5$ | $0.31 \pm 0.04$ | $0.5$ | $0.49 \pm 0.05$ | $0.5$ |
| | Population | $0.46 \pm 0.07$ | $0.03 \pm 0.04$ | $0.43 \pm 0.06$ | $0.07 \pm 0.03$ | $0.50 \pm 0.06$ | $0.03 \pm 0.02$ |
| CIFAR–2$A$ | THREAD-LinUCB | $\mathbf{0.15 \pm 0.04}$ | $\mathbf{0.31 \pm 0.02}$ | $\mathbf{0.14 \pm 0.05}$ | $\mathbf{0.32 \pm 0.02}$ | $0.34 \pm 0.07$ | $0.25 \pm 0.07$ |
| | THREAD-KNN | $0.14 \pm 0.04$ | $0.33 \pm 0.06$ | $\mathbf{0.14 \pm 0.04}$ | $\mathbf{0.32 \pm 0.03}$ | $\mathbf{0.33 \pm 0.05}$ | $\mathbf{0.18 \pm 0.03}$ |
| | Best Risk | $0.1$ | $0.33$ | $0.1$ | $0.33$ | $0.3$ | $0.17$ |

| | Algorithm | $r_{A_1} = [0.8, 0.8, 0.1, 0.1]$ $r_{A_2} = [0.2, 0.1, 0.1, 0.2]$ | | $r_{A_1} = [0.2, 0.8, 0.1, 0.1]$ $r_{A_2} = [0.8, 0.1, 0.1, 0.2]$ | | $r_{A_1} = [0.2, 0.5, 0.5, 0.8]$ $r_{A_2} = [0.8, 0.1, 0.1, 0.8]$ | |
| --- | --- | --- | --- | --- | --- | --- | --- |
| | | $r_h(\pi)$ | $c(\pi)$ | $r_h(\pi)$ | $c(\pi)$ | $r_h(\pi)$ | $c(\pi)$ |
| | H-ONLY ($\pi^*_{\lambda=0}$) | $0.45 \pm 0.04$ | $0.0$ | $0.33 \pm 0.06$ | $0.0$ | $0.51 \pm 0.04$ | $0.0$ |
| | H-MODEL | $0.14 \pm 0.02$ | $0.1$ | $0.30 \pm 0.05$ | $0.1$ | $0.45 \pm 0.04$ | $0.1$ |
| MMLU-2$A$ | Human Pop | $0.45 \pm 0.07$ | $0.05 \pm 0.01$ | $0.22 \pm 0.08$ | $0.06 \pm 0.01$ | $0.41 \pm 0.04$ | $0.05 \pm 0.01$ |
| | THREAD-LinUCB | $0.15 \pm 0.04$ | $0.05 \pm 0.06$ | $0.19 \pm 0.07$ | $0.05 \pm 0.02$ | $0.39 \pm 0.08$ | $0.06 \pm 0.01$ |
| | THREAD-KNN | $\mathbf{0.14 \pm 0.06}$ | $\mathbf{0.04 \pm 0.01}$ | $\mathbf{0.13 \pm 0.04}$ | $\mathbf{0.03 \pm 0.04}$ | $\mathbf{0.33 \pm 0.05}$ | $\mathbf{0.05 \pm 0.01}$ |
| | Best Risk ($\pi^*_{\lambda=1}$) | $0.13$ | $0.05$ | $0.13$ | $0.03$ | $0.3$ | $0.05$ |

Selected $\lambda$ Values

| | Algorithm | $r_{A_1} = [0.7, 0.1, 0.7]$ $r_{A_2} = [0.1, 0.1, 0.1]$ | $r_{A_1} = [0.7, 0.1, 0.7]$ $r_{A_2} = [0.1, 0.7, 0.1]$ | $r_{A_1} = [0.7, 0.1, 0.7]$ $r_{A_2} = [0.7, 0.7, 0.1]$ |
| --- | --- | --- | --- | --- |
| Synthetic-2$A$ | THREAD-LinUCB | $0.4$ | $0.6$ | $0.6$ |
| | THREAD-KNN | $0.9$ | $0.6$ | $0.6$ |
| CIFAR-2$A$ | THREAD-LinUCB | $0.4$ | $0.9$ | $0.5$ |
| | THREAD-KNN | $0.7$ | $0.9$ | $0.6$ |

| | Algorithm | $r_{A_1} = [0.8, 0.8, 0.1, 0.1]$ $r_{A_2} = [0.2, 0.1, 0.1, 0.2]$ | $r_{A_1} = [0.2, 0.8, 0.1, 0.1]$ $r_{A_2} = [0.8, 0.1, 0.1, 0.2]$ | $r_{A_1} = [0.2, 0.5, 0.5, 0.8]$ $r_{A_2} = [0.8, 0.1, 0.1, 0.8]$ |
| --- | --- | --- | --- | --- |
| MMLU-2$A$ | THREAD-LinUCB | $1.0$ | $0.9$ | $1.0$ |
| | THREAD-KNN | $0.3$ | $0.2$ | $0.3$ |

a variety of real-world contexts, but may come at the added cost of an increase in the number of required interactions, if we learn with no prior assumptions on the decision-maker's expertise profile.

$|\mathcal{A}| > 3$ **Experiments.** We now demonstrate how THREAD behaves when we increase the number of forms of support available to a decision-maker. Using the same set-up and algorithm-specific hyperparameters as Table 1, we convert the CIFAR setting to five forms of support (one for each class), resulting in CIFAR-5$A$, and convert the MMLU setting to four forms of support (one for each topic) resulting in MMLU-4$A$. In

Table 5: The best value of $\lambda$ that is selected from doing a sweep over $\lambda \in [0, 1]$ that corresponds to the human simulators in Table 2. The column ordering is the same as the main paper.

| | Algorithm | Best Alone | Second Best Alone | Highest Best Risk | Lowest Best Risk |
|---|---|---|---|---|---|
| MMLU-2$A$ | THREAD-LinUCB | 0.7 | 0.65 | 1.0 | 0.95 |
| | THREAD-KNN | 0.25 | 0.55 | 0.35 | 0.6 |

Table 6: Vary exploration parameters ($\alpha$ for LinUCB and $\gamma$ for KNN). We find that LinUCB generally performs better when exploration increases and KNN generally performs better when exploration decreases.

| | | $r_{A_1} = [0.7, 0.1, 0.7]$ $r_{A_2} = [0.1, 0.1, 0.1]$ | | $r_{A_1} = [0.7, 0.1, 0.7]$ $r_{A_2} = [0.1, 0.7, 0.1]$ | | $r_{A_1} = [0.7, 0.1, 0.7]$ $r_{A_2} = [0.7, 0.7, 0.1]$ | |
|---|---|---|---|---|---|---|---|
| | Algorithm | $r_h(\pi)$ | $c(\pi)$ | $r_h(\pi)$ | $c(\pi)$ | $r_h(\pi)$ | $c(\pi)$ |
| Synthetic-2$A$ | THREAD-LinUCB ($\alpha = 0.1$) | $0.14 \pm 0.04$ | $0.33 \pm 0.02$ | $0.19 \pm 0.09$ | $0.32 \pm 0.02$ | $0.34 \pm 0.10$ | $0.21 \pm 0.07$ |
| | THREAD-LinUCB ($\alpha = 10$) | $0.13 \pm 0.04$ | $0.32 \pm 0.03$ | $0.12 \pm 0.03$ | $0.33 \pm 0.02$ | $0.30 \pm 0.05$ | $0.23 \pm 0.07$ |
| | THREAD-KNN ($\gamma = 0.01$) | $0.12 \pm 0.04$ | $0.35 \pm 0.06$ | $0.19 \pm 0.08$ | $0.26 \pm 0.07$ | $0.33 \pm 0.04$ | $0.20 \pm 0.03$ |
| | THREAD-KNN ($\gamma = 0.2$) | $0.16 \pm 0.05$ | $0.33 \pm 0.04$ | $0.21 \pm 0.06$ | $0.24 \pm 0.03$ | $0.34 \pm 0.06$ | $0.15 \pm 0.02$ |
| CIFAR-2$A$ | THREAD-LinUCB ($\alpha = 0.1$) | $0.15 \pm 0.08$ | $0.30 \pm 0.04$ | $0.14 \pm 0.05$ | $0.32 \pm 0.02$ | $0.31 \pm 0.05$ | $0.26 \pm 0.07$ |
| | THREAD-LinUCB ($\alpha = 10$) | $0.14 \pm 0.04$ | $0.32 \pm 0.02$ | $0.18 \pm 0.04$ | $0.31 \pm 0.02$ | $0.32 \pm 0.05$ | $0.26 \pm 0.08$ |
| | THREAD-KNN ($\gamma = 0.01$) | $0.12 \pm 0.03$ | $0.33 \pm 0.02$ | $0.14 \pm 0.06$ | $0.31 \pm 0.03$ | $0.35 \pm 0.06$ | $0.14 \pm 0.03$ |
| | THREAD-KNN ($\gamma = 0.2$) | $0.17 \pm 0.05$ | $0.32 \pm 0.06$ | $0.20 \pm 0.09$ | $0.31 \pm 0.03$ | $0.35 \pm 0.06$ | $0.16 \pm 0.04$ |

| | | $r_{A_1} = [0.8, 0.8, 0.1, 0.1]$ $r_{A_2} = [0.2, 0.1, 0.1, 0.2]$ | | $r_{A_1} = [0.2, 0.8, 0.1, 0.1]$ $r_{A_2} = [0.8, 0.1, 0.1, 0.2]$ | | $r_{A_1} = [0.2, 0.5, 0.5, 0.8]$ $r_{A_2} = [0.8, 0.1, 0.1, 0.8]$ | |
|---|---|---|---|---|---|---|---|
| | Algorithm | $r_h(\pi)$ | $c(\pi)$ | $r_h(\pi)$ | $c(\pi)$ | $r_h(\pi)$ | $c(\pi)$ |
| MMLU-2$A$ | THREAD-LinUCB ($\alpha = 0.1$) | $0.26 \pm 0.06$ | $0.03 \pm 0.01$ | $0.22 \pm 0.09$ | $0.04 \pm 0.01$ | $0.47 \pm 0.08$ | $0.03 \pm 0.02$ |
| | THREAD-LinUCB ($\alpha = 10$) | $0.15 \pm 0.05$ | $0.04 \pm 0.01$ | $0.18 \pm 0.07$ | $0.03 \pm 0.01$ | $0.40 \pm 0.06$ | $0.05 \pm 0.01$ |
| | THREAD-KNN ($\gamma = 0.01$) | $0.16 \pm 0.05$ | $0.05 \pm 0.00$ | $0.15 \pm 0.06$ | $0.03 \pm 0.01$ | $0.39 \pm 0.06$ | $0.03 \pm 0.01$ |
| | THREAD-KNN ($\gamma = 0.2$) | $0.15 \pm 0.03$ | $0.05 \pm 0.01$ | $0.12 \pm 0.03$ | $0.03 \pm 0.00$ | $0.38 \pm 0.05$ | $0.03 \pm 0.01$ |

Table 9, we report $L_h(\pi)$ at three time steps: $t = 50, 100, 250$. For CIFAR, we find that KNN significantly outperforms given more interactions, which is expected when we increase $|\mathcal{A}|$. We believe that LinUCB struggles as the t-SNE embedding space is likely not rich enough to capture the intricacies of all five forms of support. Given a larger embedding space, we expect that LinUCB would excel when the number of forms of support is increased. For MMLU, both LinUCB and KNN perform well by the 250th interaction: KNN is very good at lowering $L_h(\pi)$ on this task. In all, this experiment assures us that careful design of decision support can permit us to use THREAD when we have a larger set of potential forms of support.

While we include experiments with larger $|\mathcal{A}|$ here, we note that higher set sizes may also be problematic for two reasons: 1) decision-makers may struggle under many forms of support (Kalis et al., 2013), and 2) as size increases, we find that the number of interactions required to learn an accurate personalized policy also increases.

**Sensitivity to choice of $T$.** In Figure 10, we consider a setting of learning a policy with three forms of support: $\mathcal{A} = \{A_1, A_2, A_3\}$. Let $c(A_1) < c(A_2) = c(A_3)$. This is an analogous setup to what we do in our human subject experiments in Section 6. We indicate the simulated human expertise profiles in the caption. We show how performance on a random sample of 100 points from a held-out set varies with time while learning a decision support policy. In general, we find that the more forms of support to learn, the longer it

Table 7: We vary the $K$ and $W$ parameters used to instantiate KNN across three datasets and report the expected risk $L_h(\pi)$ (lower is better) and expected cost $c(\pi)$ (lower is better) across 5 runs. The human simulation used here is the same as the one used in the second setting of Table 4.

| | $K = 3$, $W = 10$ | | $K = 5$, $W = 10$ | | $K = 5$, $W = 25$ | | $K = 8$, $W = 40$ | |
|---|---|---|---|---|---|---|---|---|
| | $L_h(\pi)$ | $c(\pi)$ | $L_h(\pi)$ | $c(\pi)$ | $L_h(\pi)$ | $c(\pi)$ | $L_h(\pi)$ | $c(\pi)$ |
| Synthetic-2$A$ | $0.20 \pm 0.11$ | $0.25 \pm 0.08$ | $0.15 \pm 0.05$ | $0.30 \pm 0.03$ | $0.14 \pm 0.05$ | $0.31 \pm 0.03$ | $0.12 \pm 0.04$ | $00 \pm 00$ |
| CIFAR-2$A$ | $0.25 \pm 0.15$ | $0.24 \pm 0.12$ | $0.14 \pm 0.04$ | $0.34 \pm 0.03$ | $0.17 \pm 0.04$ | $0.30 \pm 0.04$ | $0.14 \pm 0.04$ | $0.32 \pm 0.03$ |
| MMLU-2$A$ | $0.18 \pm 0.07$ | $0.03 \pm 0.01$ | $0.14 \pm 0.04$ | $0.03 \pm 0.01$ | $0.15 \pm 0.03$ | $0.04 \pm 0.01$ | $0.15 \pm 0.04$ | $0.03 \pm 0.01$ |

Table 8: On the CIFAR-2$A$ dataset, we explore the effect of varying the embedding size for KNN, with $K = 8$, $W = 25$ (Top), and LinUCB (Bottom). We report the expected risk $L_h(\pi)$ (lower is better) and expected cost $c(\pi)$ (lower is better) averaged over 5 runs. Recall that in our main paper experiments we used two-dimensional t-SNE embeddings, not these model embeddings, for ease of visualization.

| Embedding Size | $r_{A_1} = [0.7, 0.1, 0.7]$ $r_{A_2} = [0.1, 0.7, 0.1]$ | | | | $r_{A_1} = [0.7, 0.1, 0.7]$ $r_{A_2} = [0.7, 0.7, 0.1]$ | | | |
|---|---|---|---|---|---|---|---|---|
| | $T = 100$ | | $T = 200$ | | $T = 100$ | | $T = 200$ | |
| | $L_h(\pi)$ | $c(\pi)$ | $L_h(\pi)$ | $c(\pi)$ | $L_h(\pi)$ | $c(\pi)$ | $L_h(\pi)$ | $c(\pi)$ |
| 2 | $0.18 \pm 0.08$ | $0.28 \pm 0.06$ | $0.14 \pm 0.04$ | $0.32 \pm 0.03$ | $0.32 \pm 0.05$ | $0.24 \pm 0.06$ | $0.33 \pm 0.06$ | $0.23 \pm 0.05$ |
| 4 | $0.18 \pm 0.04$ | $0.29 \pm 0.04$ | $0.16 \pm 0.04$ | $0.32 \pm 0.03$ | $0.33 \pm 0.05$ | $0.20 \pm 0.05$ | $0.33 \pm 0.05$ | $0.21 \pm 0.04$ |
| 8 | $0.15 \pm 0.05$ | $0.30 \pm 0.04$ | $0.15 \pm 0.03$ | $0.33 \pm 0.03$ | $0.32 \pm 0.05$ | $0.24 \pm 0.04$ | $0.33 \pm 0.05$ | $0.24 \pm 0.06$ |
| 512 | $0.15 \pm 0.05$ | $0.30 \pm 0.04$ | $0.15 \pm 0.03$ | $0.33 \pm 0.03$ | $0.35 \pm 0.05$ | $0.24 \pm 0.07$ | $0.35 \pm 0.05$ | $0.23 \pm 0.04$ |

| Embedding Size | $r_{A_1} = [0.7, 0.1, 0.7]$ $r_{A_2} = [0.1, 0.7, 0.1]$ | | | | $r_{A_1} = [0.7, 0.1, 0.7]$ $r_{A_2} = [0.7, 0.7, 0.1]$ | | | |
|---|---|---|---|---|---|---|---|---|
| | $T = 100$ | | $T = 200$ | | $T = 100$ | | $T = 200$ | |
| | $L_h(\pi)$ | $c(\pi)$ | $L_h(\pi)$ | $c(\pi)$ | $L_h(\pi)$ | $c(\pi)$ | $L_h(\pi)$ | $c(\pi)$ |
| 2 | $0.31 \pm 0.05$ | $0.19 \pm 0.03$ | $0.32 \pm 0.04$ | $0.20 \pm 0.02$ | $0.49 \pm 0.05$ | $0.10 \pm 0.08$ | $0.49 \pm 0.05$ | $0.11 \pm 0.08$ |
| 4 | $0.14 \pm 0.04$ | $0.36 \pm 0.02$ | $0.14 \pm 0.04$ | $0.36 \pm 0.02$ | $0.32 \pm 0.04$ | $0.23 \pm 0.06$ | $0.31 \pm 0.50$ | $0.22 \pm 0.05$ |
| 8 | $0.13 \pm 0.04$ | $0.35 \pm 0.02$ | $0.13 \pm 0.03$ | $0.36 \pm 0.02$ | $0.31 \pm 0.03$ | $0.20 \pm 0.03$ | $0.31 \pm 0.04$ | $0.20 \pm 0.04$ |
| 512 | $0.17 \pm 0.03$ | $0.31 \pm 0.03$ | $0.16 \pm 0.04$ | $0.31 \pm 0.03$ | $0.34 \pm 0.04$ | $0.24 \pm 0.05$ | $0.33 \pm 0.06$ | $0.25 \pm 0.04$ |

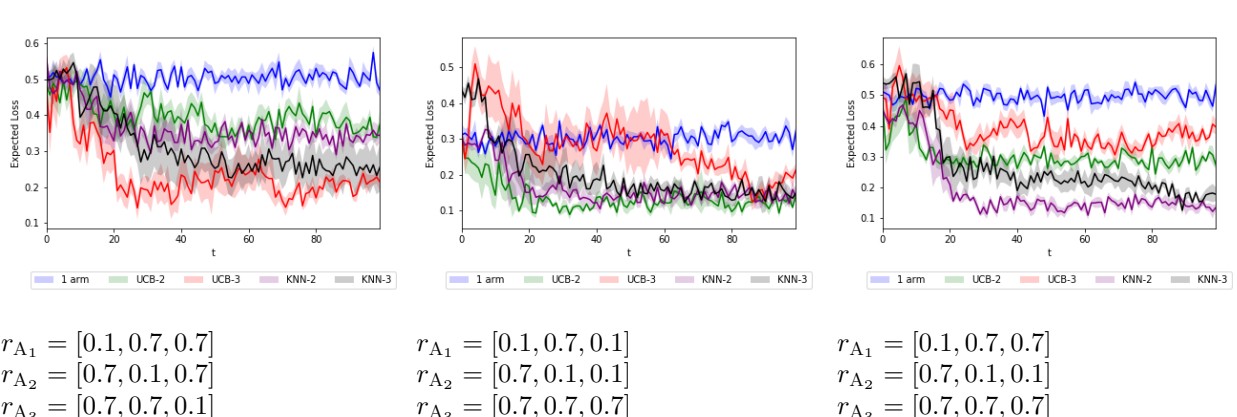

$r_{A_1} = [0.1, 0.7, 0.7]$
$r_{A_2} = [0.7, 0.1, 0.7]$
$r_{A_3} = [0.7, 0.7, 0.1]$

$r_{A_1} = [0.1, 0.7, 0.1]$
$r_{A_2} = [0.7, 0.1, 0.1]$
$r_{A_3} = [0.7, 0.7, 0.7]$

$r_{A_1} = [0.1, 0.7, 0.7]$
$r_{A_2} = [0.7, 0.1, 0.1]$
$r_{A_3} = [0.7, 0.7, 0.7]$

Figure 10: Loss plots over time for human simulations (denoted by $r_{A_i}$) in the CIFAR-3$A$ setting.

may take for the online algorithms to "converge." For various simulations, KNN is more consistent in learning a decision support policy efficiently.

Table 9: We evaluate $|\mathcal{A}| > 3$ in the cost-agnostic setting, $L_h(\pi_t)$ for $t = 50, 100, 250$ for the same setup as Table 1, but with more arms. We find that THREAD works well over time. The success of LinUCB is highly dependent on the embedding space and its geometry.

| | Algorithm | $T = 50$ | $T = 100$ | $T = 250$ |
|---|---|---|---|---|
| CIFAR-5$A$ | THREAD-LinUCB | $0.59 \pm 0.10$ | $0.29 \pm 0.17$ | $0.34 \pm 0.10$ |
| | THREAD-KNN | $0.19 \pm 0.15$ | $0.14 \pm 0.17$ | $0.11 \pm 0.15$ |
| MMLU-4$A$ | THREAD-LinUCB | $0.13 \pm 0.10$ | $0.10 \pm 0.07$ | $0.17 \pm 0.09$ |
| | THREAD-KNN | $0.11 \pm 0.12$ | $0.08 \pm 0.12$ | $0.02 \pm 0.04$ |

# E Additional User Study Details and Results

## E.1 Additional Recruitment Details

Within a task, participants are randomly assigned to an algorithm variant; an equal number of participants are included per variant (i.e., 10 for MMLU and 5 for CIFAR)[9]. Participants are required to reside in the United States and speak English as a first-language. Participants are paid at a base rate of \$9/hr, and are told they may be paid an optional bonus up to \$10/hr based on the number of correct responses. We allot 25-30 minutes for the CIFAR task and 30-40 for MMLU, as each MMLU question takes more effort. We applied the bonus to all participants.

### E.1.1 CIFAR Algorithm Parameters

For KNN, we take $K = 8$, $W = 25$, and $\epsilon = 0.1$; for both algorithms, we let $\alpha = 1$. Cost of either model arm is 0.5. For each algorithm, we consider two different settings of $\lambda$ ($\lambda \in \{0.75, 1.0\}$ and $\in \{0.85, 1.0\}$ for KNN and LinUCB respective), selected using the hyper-parameter selection methods per Section 3.2.

### E.1.2 MMLU Algorithm Parameters

We use the same parameters as in the CIFAR task, with the exception that we set the cost of support (i.e., the LLM arm) to 0.1. Additionally, we employ different $\lambda$ – $\lambda \in \{0.75, 1.0\}$ for KNN and $\lambda \in \{0.95, 1.0\}$ for LinUCB – selected using the hyper-parameter selection methods per Section 3.2.

## E.2 Additional HSE Results

**Visualizing Learned Policies.** We provide additional details on how the policy generalization snapshots in Figures 3 and 4 are constructed. We save out the parameters learned by each algorithm while a participant interacts with Modiste. After the 60 or 100 trials (depending on which task they participated in), we load in the final state of parameters. We sample the embeddings of *unseen points* from the respective task dataset (as described in Appendix C) and pass these through the respective algorithm (LinUCB or KNN, depending on which variant the participant was assigned to) – yielding a recommended form of support for said embedding. We color the latent by the form of support. We draw randomly 250 such unseen examples for CIFAR, and 100 for MMLU, when computing the policy recommendations.

We include snapshots for participants, across both tasks, in tha main text. We depict all CIFAR examples and a random sampling of 7 of the 10 participants per variant for MMLU.

**Cumulative results.** We also report in Table 10 the expected loss and cost across all trials. In general, we find similar trends as Table 3.

Table 10: We report expected loss $L_h(\pi)$ and expected cost $c(\pi)$ incurred (lower is better) across *all* trials by Prolific participants for each Algorithm and **bold** the variant with the lowest $L_h(\pi)$. Modiste learns effective, low-cost policies: for CIFAR, Modiste outperforms all baselines. For MMLU, we find that Modiste learns a policy with roughly the same performance as the best offline policy but at *half* the cost. We also consider different choices of $\lambda$, where $\lambda = 1.0$ corresponds to the standard setting and $\lambda \neq 1.0$ corresponds to a cost-aware setting, where the choice of $\lambda$ was selected according to Section 3.2.

| Algorithm | $L_h(\pi)$ | $c(\pi)$ |
|---|---|---|
| H-ONLY | $0.57 \pm 0.06$ | 0 |
| H-MODEL | $0.44 \pm 0.04$ | 0.5 |
| H-CONSENSUS | $0.32 \pm 0.06$ | 0.5 |
| Population | $0.6 \pm 0.02$ | 0 |
| Modiste (LinUCB, $\lambda = 1.0$) | $0.28 \pm 0.03$ | $0.35 \pm 0.01$ |
| Modiste (KNN, $\lambda = 1.0$) | $\mathbf{0.24 \pm 0.03}$ | $\mathbf{0.39 \pm 0.04}$ |
| Modiste (LinUCB, $\lambda = 0.85$) | $0.37 \pm 0.07$ | $0.25 \pm 0.01$ |
| Modiste (KNN, $\lambda = 0.75$) | $0.24 \pm 0.03$ | $0.4 \pm 0.03$ |

| Algorithm | $L_h(\pi)$ | $c(\pi)$ |
|---|---|---|
| H-ONLY | $0.44 \pm 0.07$ | 0 |
| H-LLM | $\mathbf{0.22 \pm 0.06}$ | $\mathbf{0.1}$ |
| Population | $0.34 \pm 0.08$ | $0.05 \pm 0.00$ |
| Modiste (LinUCB, $\lambda = 1.0$) | $0.26 \pm 0.07$ | $0.05 \pm 0.00$ |
| Modiste (KNN, $\lambda = 1.0$) | $0.32 \pm 0.1$ | $0.06 \pm 0.01$ |
| Modiste (LinUCB, $\lambda = 0.95$) | $0.31 \pm 0.09$ | $0.06 \pm 0.01$ |
| Modiste (KNN, $\lambda = 0.75$) | $0.32 \pm 0.1$ | $0.06 \pm 0.01$ |

[9]Due to a server-side glitch, 6 of the 125 recruited participants received incorrect feedback on $\leq 2\%$ of trials.

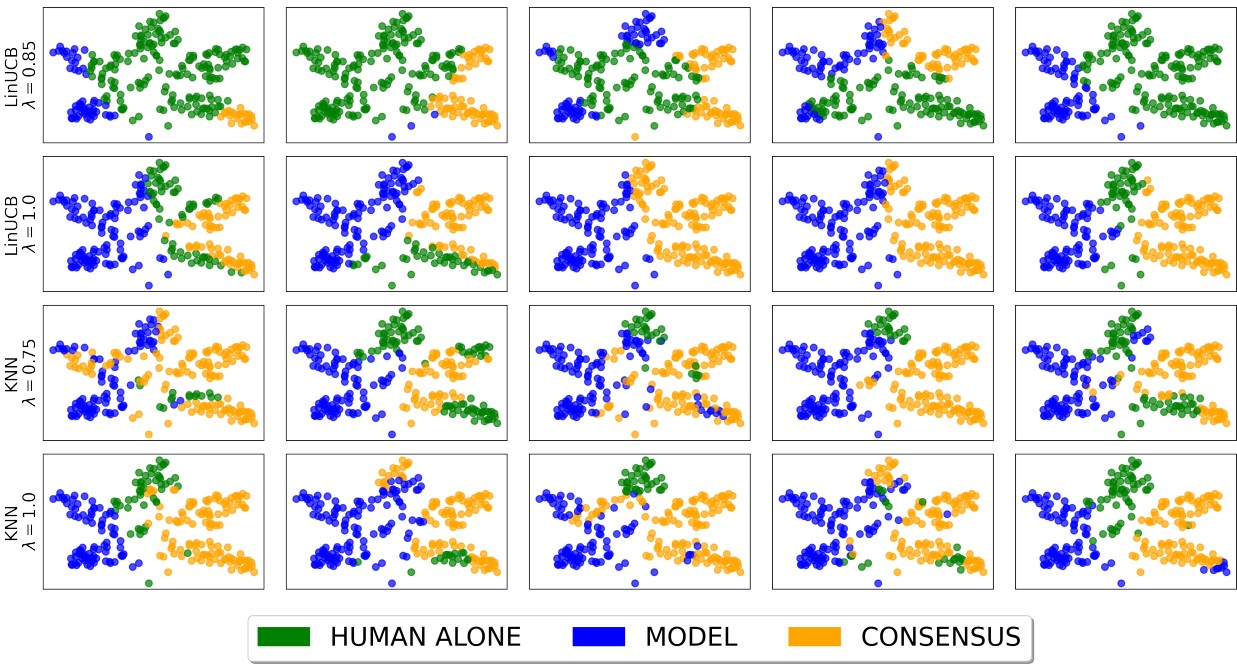

Figure 11: Snapshots of the recommended forms of support learned via `Modiste` for all participants in the CIFAR task, for different $\lambda$. Policies learned using KNN get closer to optimal; see Figure 3.

# F   HSE Discussion

**Participant Comments Hint at Mental Models of AI**

We observed that several participants in our MMLU task noted[10] that their responses were biased by the AI. We include a sampling of comments provided by participants which we found particularly revealing. We believe that these responses motivate the need for further study of how users' mental models of AI systems inform their decision-making performance, particularly in light of the rapidly advancing and public-facing foundation models (Bommasani et al., 2021).

- "I thought the AI answers went against my previous notions about what an AI
  would be proficient at answering.  I thought an AI would excel at anything
  where there is a set answer such as math, computer science coding question, and
  biology textbook definitions.  Then I expected it to absoutly fail at political,
  and historical questions.  Since they can be more opinionated and not have a
  definitive answer.  However, the AI struggled with math and excelled in the other
  topics.  So I was actively not choosing to agree with the AI on any math where I
  could also easily check its work and was heavily relying on it for everything else
  [...]  I knew nothing about any question on computer science, so I mainly relied
  on AI for answers on that topic."

- "AI seemed to me more helpful in topics I didn't know, but sometime I trusted it
  too much, instead of thinking the answer through"

- "The AI had me second guessing myself sometimes so it affected my answers
  occasionally."

- "Anything highlighted with the AI I was confident that would be the right answer."

---

[10]We included a post-experiment survey where we permitted participants to leave general comments.

