# OpenReview forum: "Learning Personalized Decision Support Policies"
_TMLR — Rejected by TMLR_

### Review · Reviewer_7xJC · 2023-10-22

**Summary Of Contributions:**

This paper formulates a multiclass classification task given some form of support to the decision maker. It proposes to formulate the choice of support forms as a contextualized bandits where the goal is to find the optimal form of support given the context. It proposes to use LinUCB and KNN-UCB to solve the problem in an online setting. Extensive experiments demonstrated the effectiveness of these algorithms in solving this problem in terms of prediction loss and support costs.

**Audience:**

No

**Broader Impact Concerns:**

Not needed.

**Claims And Evidence:**

No

**Requested Changes:**

1. Please discuss the optimality of your algorithm in this setting by providing a (approximate) lower bound for the loss or costs.
2. Please elaborate on Appendix B.1, the proof steps should be explained in detail.
3. Please elaborate on Appendix B.1.2., this appendix looks like just a claim and there is no supporting proofs or reasoning. If you prefer to leave it at that, why does this appendix even exist?
4. Please discuss the optimality of the hyperparameter tuning approach proposed for tuning $\lambda$. How could you connect this to some part of the vast literature of multi-objective optimization?

**Strengths And Weaknesses:**

Strengths:
1. The paper conducts extensive experiments with publicly available data. It also includes crowdsourced experiments with human evaluators. This provides pretty unique insights in terms of the effectiveness of the algorithms.

 Weaknesses:
1. The paper uses existing algorithms and does not include any forms of technical novelty. The algorithms used for solving the problem are outdated and probably sub-optimal for this setting.
2. The proofs in the appendix need major re-write and clarification. I added more details in "Requested Changes".
3. The procedure in Section 3.3. does not scale very well for larger numbers of regions $\mathcal{X}\_j $ or support forms $A_i$ as it tries to estimate an arbitrary function, $r\_{A_i}( \mathcal{X}\_j, h)$ for all pairs of $\mathcal{X}\_j$ and $A_i$.
4. The hyperparameter tuning for $\lambda$ in Section 3.2 is basically a grid search. We should wonder if we need this hyperparameter tuning really or it is just problem dependent, and we should leave it to the user. If we need this tuning, why should we use such a suboptimal and practically infeasible approach like grid search.

---

> ### Author Response · Authors · 2023-11-16
> **Response to Reviewer 7xJC**
>
> We thank the reviewer for their helpful feedback. We respond to the requested changes below:
>
> *"1. Please discuss the optimality of your algorithm in this setting by providing a (approximate) lower bound for the loss or costs"* Please refer to the general response.
>
> *"2. Please elaborate on Appendix B.1, the proof steps should be explained in detail."* We have extended the text in Appendix B.1 appropriately.
>
> *"3. Please elaborate on Appendix B.1.2., this appendix looks like just a claim and there is no supporting proofs or reasoning."* Thanks for pointing this out! We have added more context in Appendix B.1.2.
>
> *"4. Please discuss the optimality of the hyperparameter tuning approach proposed for tuning \lambda. How could you connect this to some part of the vast literature of multi-objective optimization?"* Please refer to our general response on novelty.  Additionally, in the submission, we mentioned some prior work on the multi-objective optimization literature in the last paragraph of Section 2.

---

### Review · Reviewer_D96s · 2023-10-30

**Summary Of Contributions:**

This paper studies on the problem of how to learn a personalized decision policy online, i.e., which form of support should the system provide to the user for better assistance for the task. It formulates the problem as an online learning problem, and proposes THREAD, which utilizes online learning techniques, such as LinUCB to learn the optimal policy. Beyond this, it evaluates the effectiveness of the algorithm on real-world human subject experiments, building tools Modiste for an interactive test of the algorithms.

**Audience:**

No

**Claims And Evidence:**

Yes

**Requested Changes:**

- The technical contributions are not clear, as it seems direct applications of LinUCB and knnUCB. Could the authors comment the specific challenges they are facing when applying bandit algorithms in learning support policies?

- The hyper-parameter selection part needs re-written in a clear way, with theoretical support.

- It would be great to perform experiments (at least synthetically) for large action spaces.

**Strengths And Weaknesses:**

Strength:

- This paper proposes an interesting and practical setup about learning a decision support policy online, which studies the effective way of utilizing various supports to facilitate decision making, even under cost constraint.
- The paper introduces an effective algorithm THREAD, which builds upon online learning algorithms, such as LinUCB and knnUCB for learning the optimal policy.
- The human subject experiments are encouraging, validating the effectiveness of THREAD.

Weakness:

- One thing I am concerned is the technical novelty of the paper and whether TMLR is a good place for the paper. Though the paper proposes a new setup, it is just one instance of the classical bandit problem, which context being the features of the user, action being the supports being utilized, and the reward is measured by the loss of the user's prediction accuracy while using this kind of support. The algorithms are direct application of LinUCB and knnUCB. The cost-aware setup is also being discussed in the previous bandit literature already. From the technical side, the novelty is low.

- The algorithm highlights its hyper-parameter tuning under cost-aware setup. I found two paragraphs in Section 3.2 is very confusing. For example, given the policy learned under each $\lambda$, how to evaluate it effectively? Is there any theoretical support of utilizing a population-level $\lambda$ for any new user? what are the underlying assumptions here? It would be great if the authors could re-visit this part.

- The human subject experiments is pretty nice, however, all the setups are considering small action spaces, with $k=2,3$, how the method works under large action spaces?

---

> ### Author Response · Authors · 2023-11-16
> **Response to Reviewer D96s**
>
> We thank the reviewer for their constructive feedback. We respond to the requested changes below:
>
> *“The technical contributions are not clear, as it seems direct applications of LinUCB and knnUCB. Could the authors comment the specific challenges they are facing when applying bandit algorithms in learning support policies?”* Please refer to our general response. Thanks for the suggestion to discuss challenges! In the limitations paragraph of Section 6.4, we have added examples of the practical challenges encountered when implementing bandit algorithms. In summary, the challenges included tuning the explore-exploit tradeoff, which did not matter much in practice, and selecting a reasonable action space.
>
> *"The hyper-parameter selection part needs re-written in a clear way, with theoretical support."* Thank you for your suggestion. Please refer to our general response.
>
> *"It would be great to perform experiments (at least synthetically) for large action spaces."* As discussed in Section 4.1, our experiments with a small set of actions already capture many real-world use cases. Furthermore, for our human subject experiments, a larger action space may not be cost-effective or reasonable for crowd workers to complete in the standard amount of time since it would take more iterations for the policy to converge. In Table 9, we include synthetic experiments for $|\mathcal{A}| >3$, which affirms that a larger $T$ would be necessary. We imagine that future work can propose new methods (e.g., by leveraging prior information) to improve this efficiency.

---

### Review · Reviewer_jS8W · 2023-11-06

**Summary Of Contributions:**

This paper propose a learning framework that learns a personalized support policy that chooses a decision support for given input. The proposes to use contextual bandit to learn the decision policy. Experiments validates its effectiveness.

**Audience:**

Yes

**Broader Impact Concerns:**

None.

**Claims And Evidence:**

Yes

**Requested Changes:**

See the weakness.

**Strengths And Weaknesses:**

Strengths:
1.The motivation that using contextual bandits to learn a personalized support policy for classification models are reasonable.
2.Experiments validates the effectiveness of this method.

Weakness:
1.The major concern of the paper is that the paper use "decision maker" to present the notion of just classification models. I think the term "decision maker" is usually used for reinforcement learning or generative models(LLM).
2.The paper does not well motivate why we need to learn a support for classification models. A good motivation may be a motivating example to demonstrate the effectiveness of the support for for classification models.
3.Thread only uses linear contextual bandits models, which is not suitable for practical problems as there are many nonlinear reward functions in practical problem. I suggest the authors discuss the limitation of this point.
4.The paper does not compare the training time of these methods.
5.I suggest that the paper adds a real example of decision support. Otherwise it is quite hard to follow due to lack of the background.

---

> ### Author Response · Authors · 2023-11-16
> **Response to Reviewer jS8W**
>
> We thank the reviewer for their feedback and respond to each of the weaknesses:
>
> *"1.The major concern of the paper is that the paper use "decision maker" to present the notion of just classification models. I think the term "decision maker" is usually used for reinforcement learning or generative models(LLM)."* While we agree that the focus of our work is on multi-class classification tasks, we believe that it is not a restrictive assumption as it reflects many prior studies in the human-AI decision-making literature (see Lai et al. 2023). We would be excited for future work to extend the problem formulation and conduct experiments over a broader range of settings, including the ones suggested by the reviewer.
>
> *"2.The paper does not well motivate why we need to learn a support for classification models."* Thanks for the suggestion! In the revision, we have added a clarification to the beginning of Section 3 when the problem formulation is introduced. Our human subject experiments for two different classification tasks (CIFAR and MMLU) are both demonstrations of how decision-makers benefit from different forms of support.
>
> *"3.Thread only uses linear contextual bandits models, which is not suitable for practical problems."* We respectfully disagree with this comment for two reasons. First, LinUCB is commonly used in many practical problems like ad recommendation (see Li et al. 2010). Second, one of the bandit methods considered in our work is based on KNN, which is not a linear approach.
>
> *"4.The paper does not compare the training time of these methods. Also, the paper does not discussion its limitations."* Since we implement existing online learning algorithms, where the update steps for both LinUCB and KNN are relatively simple, we do not believe there is a relevant notion of “training time” to report. We highlight that THREAD was running in real-time during our human subject experiments (Table 3), which further emphasizes the practical usefulness of our proposed approach. While we mentioned some limitations of this work in the last paragraph of the conclusion in the submitted version, we have highlighted this discussion more prominently in Section 6.4 of the revision.
>
> *"5.I suggest that the paper adds an real example of decision support. Otherwise it is quite hard to follow due to lack of the background."* In the submission, the introduction and Figure 1 illustrate one example where a radiologist may be provided different forms of support (e.g., ML prediction, LLM summary, or no support) to enable a better diagnosis. Our human subject experiments also provide two more concrete examples of real decision support with real human participants. If we have misunderstood what the reviewer is referring to as “a real example of decision support", please let us know and we would be happy to follow up.

---

### Author Response · Authors · 2023-11-16
**General response on novelty**

We would like to thank all reviewers for their invaluable feedback. We appreciate their efforts to strengthen our work. We respond to related comments from Reviewers D96s and 7xJC in a general post:

*“One thing I am concerned is the technical novelty of the paper.” “The algorithms used for solving the problem are outdated and probably sub-optimal for this setting.”* We agree that our submission does not propose a novel online learning algorithm and the algorithms we use may not be state-of-the-art in the literature. Rather, we devise a new setting in which the existing online learning methodology allows us to deploy decision support policies successfully. We attempted to make this point clear throughout the paper using language like THREAD “leverages techniques from the stochastic contextual bandit literature”. **Our main claim is that "personalizing when to show decision support (e.g., GPT, expert consensus) can improve decision-maker performance.”** We hope we provided sufficient evidence for this claim (a la #2 https://jmlr.org/tmlr/acceptance-criteria.html) and built a tool, Modiste, for human subject experimentation of learning when to provide decision support policies.

*“Discuss the optimality of the hyperparameter tuning approach.”* Additionally, we do not view the hyperparameter tuning approach as a major source of innovation in this work. We evaluated one approach of selecting $\lambda$, based on a grid-search-like method, and found that it performed well empirically in our human subject validation. We invite future work to study optimal hyperparameter tuning approaches that can be effectively implemented in practice.

As for *“whether TMLR is a good place for the paper”*– we believe our contributions on learning personalized policies for decision support will be of much interest to the burgeoning human-AI team subcommunity of TMLR (e.g., https://openreview.net/forum?id=hjDYJUn9l1, https://openreview.net/forum?id=y4CGF1A8VG, https://openreview.net/forum?id=5rq8iRzHAQ).

---

### Decision · Action_Editor_Cuky · 2023-12-17

**Recommendation:** Reject

**Comment:**

See my responses to *Claims And Evidence* and *Audience*.

**Audience:**

This paper will have audience among bandit practitioners. However, the entry point for building on it is too high because it requires running a human study. To increase audience and make the paper more attractive, we suggest that you share your datasets with the community. Try to collect them in a way that permits evaluation of other policies than those that collected them. Something like this would allow others to build on your work using off-policy evaluation or optimization methods, for instance.

**Claims And Evidence:**

This paper proposes a framework for learning personalized decision support policies using LinUCB and K-nearest neighbors in the history of the agent. The goal is to learn how to aid a human labeler as a function of the labeled data point and their past responses. The proposed approach is evaluated on both synthetic and human experiments.

The proposed approach is evaluated in an actual human study, which differentiates it from most machine learning papers. The shortcoming of the study is that the setting is very simple: only 2 or 3 arms. Therefore, the evidence is only partial. Let me compare this work to a classic paper on the topic

* [Automated handwashing assistance for persons with dementia using video and
a partially observable Markov decision process](https://cs.brown.edu/courses/csci2951-k/papers/hoey08.pdf)

This paper considers 4 actions, which are more complex because they include an actual human intervention, and was written more than 10 years ago. To address this concern, we have two suggestions:

* Consider more, and more complex, actions.

* Consider more human-aided tasks that would demonstrate the general utility of the approach beyond machine-aided labeling.

**Resubmission Of Major Revision:**

The authors may consider submitting a major revision at a later time.